# Acute aerobic exercise intensity does not modulate pain potentially due to differences in fitness levels and sex effects: results from a pharmacological fMRI study

**Janne Ina Nold\*, Tahmine Fadai, Christian Büchel**

Department of Systems Neuroscience, University Medical Centre Hamburg-Eppendorf, Hamburg, Germany

## eLife Assessment

In this **valuable** study, Nold et al. examined exercise-induced pain modulation in a pharmacological within-subject fMRI study using the opioid-antagonist naloxone and different levels of aerobic exercise intensity and pain. This investigation provides **solid** evidence to show that the intensity of exercise does not seem to impact the hypoalgesic effect. Moreover, exploratory analysis identified that fitness level and sex may potentially play a role in exercise-induced hypoalgesia, and that further confirmatory studies are required in order to verify these findings.

**Abstract** Exercise might lead to a release of endogenous opioids, potentially resulting in pain relief. However, the neurobiological underpinnings of this effect remain unclear. Using a pharmacological within-subject functional magnetic resonance imaging (fMRI) study with the opioid antagonist naloxone and different levels of aerobic exercise and pain, we investigated exercise-induced hypoalgesia ($N$ = 39, 21 female). Overall, high-intensity (HI) aerobic exercise did not reduce pain as compared to low-intensity aerobic exercise. Accordingly, we observed no significant changes in the descending pain modulatory system. The μ-opioid antagonist naloxone significantly increased overall pain ratings but showed no interaction with exercise intensity. An exploratory analysis suggested an influence of fitness level (as indicated by the functional threshold power) and sex, where males showed greater hypoalgesia after HI exercise with increasing fitness levels. This effect was attenuated by naloxone and mirrored by fMRI signal changes in the medial frontal cortex, where activation also varied with fitness level and sex, and was reversed by naloxone. These results indicate that different aerobic exercise intensities have no differential effect on pain in a mixed population sample, but individual factors such as fitness level and sex might play a role. The current study underscores the need for personalised exercise interventions to enhance pain relief in healthy as well as chronic pain populations, taking into account the sex and fitness status as well as the necessity to further investigate the opioidergic involvement in exercise-induced pain modulation.

## Introduction

Exercise has been suggested to release endogenous opioids (*Koltyn, 2000*) resulting in an attenuation of pain, termed exercise-induced hypoalgesia. However, the underlying neural (*Vaegter and Jones, 2020*; *Wu et al., 2022*) and pharmacological mechanisms remain controversial (*Koltyn, 2000*;

**\*For correspondence:**
j.nold@uke.de

**eLife digest** Many people turn to exercise as a way to relieve pain, hoping it will help them feel better. One reason this might work is because exercise can release natural chemicals in the body, called endogenous opioids, which help reduce pain. However, scientists still do not fully understand how this process works.

Nold et al. explored how different levels of aerobic exercise – such as low vs. high intensity – affect how people feel pain. They used brain imaging and a medication called naloxone, which blocks the body's opioid system, to better understand what is happening in the brain during exercise. The study included 39 healthy adults and looked at how factors like fitness level and sex might influence the effects of exercise on pain and how participants perceived pain.

To determine whether high-intensity exercise provides more pain relief than low-intensity exercise, Nold et al. studied 18 males and 21 females during both high- and low-intensity exercises. Following the workout, magnetic resonance imaging was used to study brain activity as the participants received nine painful heat and pressure stimuli. Throughout the entire experiment, participants received a constant dose of either naloxone or a saline solution as a control.

The study found that high-intensity exercise did not reduce pain any more than low-intensity exercise in the overall group. There were no major changes in how pain was processed in the brain and blocking the body's opioids with naloxone made pain feel worse, regardless of how hard the participants exercised. However, more detailed analyses revealed that males with higher fitness levels experienced more pain relief after intense exercise than females. However, this effect disappeared when naloxone was given. Brain scans showed this was linked to activity in a part of the brain called the medial frontal cortex.

These findings suggest that exercise may help reduce pain for some people more than others – especially depending on their sex and fitness level. In the future, personalized exercise programs could be developed to help manage pain more effectively. But before that can happen, more research is needed to understand exactly how the body's natural pain-relief systems work during exercise, and how they differ between individuals.

*Naugle et al., 2012*; *Wewege and Jones, 2021*) and results on the modulation of pain through exercise are heterogeneous (*Klich et al., 2018*; *Koltyn et al., 2014*; *Padawer and Levine, 1992*; *Ruble et al., 2005*; *Sternberg et al., 2001*; *Vaegter et al., 2015*; *Vaegter et al., 2019*). Given that exercise can potentially elicit hypo- and hyperalgesia (*Vaegter and Jones, 2020*), we have selected the term 'exercise-induced pain modulation' to reflect the range of effects of exercise on pain.

Previous rodent research has implicated the descending pain modulatory system, including the anterior cingulate cortex (ACC), the medial frontal cortex (mFC) including the ventromedial prefrontal cortex (vmPFC), and the periaqueductal grey (PAG) in exercise-induced pain modulation (*Lesnak and Sluka, 2020*; *Stagg et al., 2011*). This is partly congruent with early imaging studies in humans (*Boecker et al., 2008*; *Ellingson et al., 2016*; *Geisler et al., 2020*; *Saanijoki et al., 2018*; *Scheef et al., 2012*), where the descending pain modulatory system, including the PAG and insula, seems to be involved in modulating pain perception following a 2-hr run (*Scheef et al., 2012*) along with the fusiform gyrus and hippocampus (*Geisler et al., 2020*).

The activation of the endogenous opioid system during exercise has been the most widely discussed mechanism in exercise-induced pain modulation (*Naugle et al., 2012*; *Sluka et al., 2018*). Studies frequently employ the μ-opioid antagonist naloxone which blocks opioid receptors in the brain, thus counteracting opioid-driven mechanisms of pain modulation. Earlier studies found an analgesic response to thermal and ischaemic pain after long-distance (6.3 miles) running in male runners that was reversed after naloxone administration (*Janal et al., 1984*). Even after a short-distance (1 mile) run, naloxone slowed the return of the analgesic effect (*Haier et al., 1981*). Previous imaging studies (*Boecker et al., 2008*; *Saanijoki et al., 2018*) further supported that exercise elicits the release of endogenous opioids in crucial opioidergic structures (i.e. the insula, ACC, and prefrontal regions). Still, the pharmacological underpinnings in humans have remained equivocal (*Koltyn, 2000*), where administering a μ-opioid antagonist in some studies decreased exercise-induced pain modulation (*Droste et al., 1988*; *Haier et al., 1981*; *Janal et al., 1984*; *Olausson et al., 1986*) whereas in

others it did not affect exercise-induced pain modulation (*Droste et al., 1991*; *Koltyn et al., 2014*; *Olausson et al., 1986*). Notably, most studies investigating the pharmacological underpinnings of exercise-induced pain modulation have employed very different interventions using different administration protocols (i.e. different doses and different routes of administration of the μ-opioid receptor antagonist such as oral or intravenous), making it difficult to draw clear conclusions about the extent of central μ-opioid modulation through exercise in humans (*Koltyn, 2000*; *Leknes and Atlas, 2020*; *Wewege and Jones, 2021*).

In human studies, many findings on exercise-induced pain modulation are derived from studies in athletes or physically active individuals showing increased hypoalgesia following exercise compared to subjects with lower fitness levels (*Crombie et al., 2018*; *Geva and Defrin, 2013*; *Naugle and Riley, 2014*; *Schmitt et al., 2020*). Intriguingly, this observation might be related to greater exercise-induced opioid release in structures such as the PAG or the frontal medial cortex (including the ACC) in fitter individuals (*Saanijoki et al., 2022*; *Sluka et al., 2018*). Additionally, previous reviews have further emphasised sex differences in pain processing (*Mogil, 2020*; *Mogil, 2012*) and analgesic sensitivity in rodents (*Martin et al., 2019*; *Mogil, 2020*) warranting further investigation in humans (*Mogil, 2020*; *Mogil, 2012*) in connection with exercise-induced pain modulation.

Furthermore, studies on exercise-induced pain modulation have been criticised (*Padawer and Levine, 1992*) for using single-arm pre–post measurements where pain is measured once before and once after the exercise intervention instead of randomised trials with the former not accounting for habituation to the noxious stimulus (for reviews see *Naugle et al., 2012*; *Vaegter and Jones, 2020*),

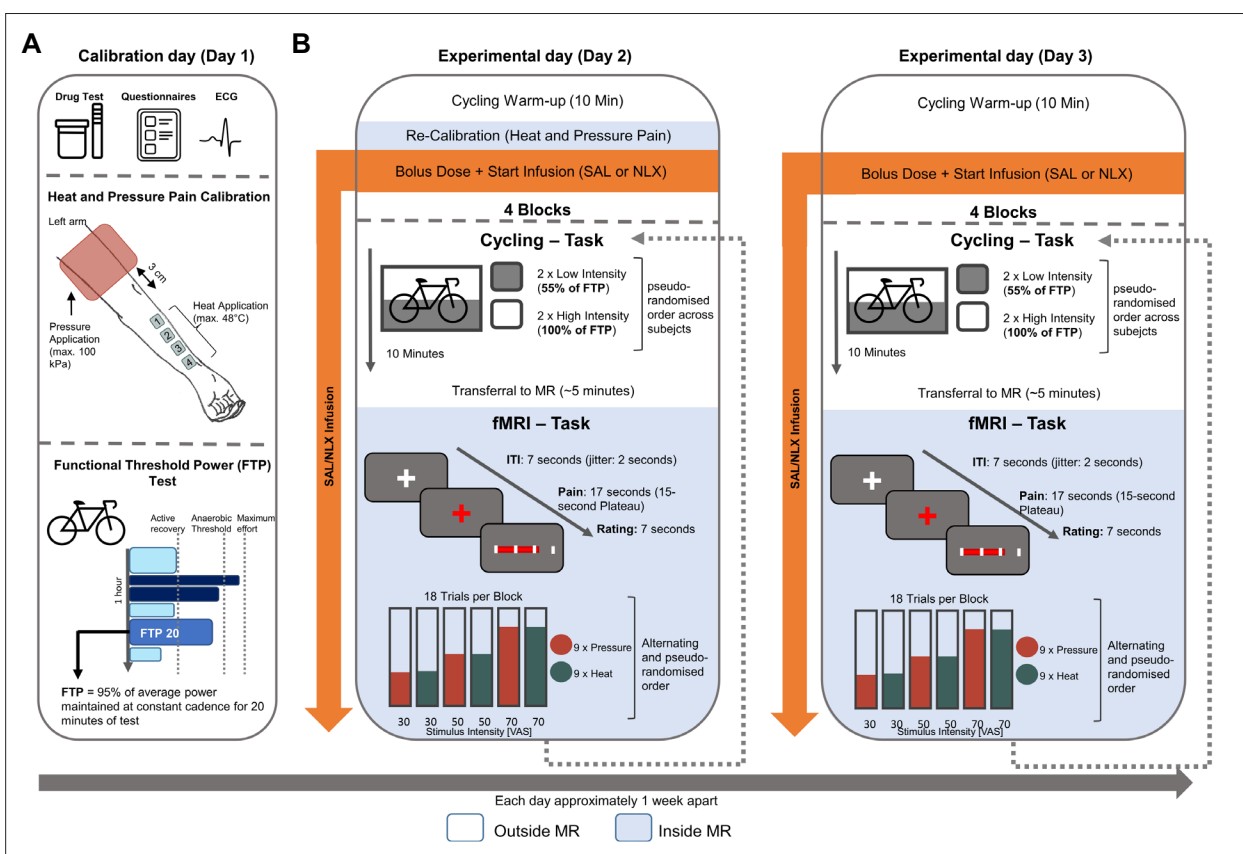

**Figure 1.** Experimental design. (**A**) Calibration day (Day 1) with heat and pressure pain calibration and functional threshold power (FTP) test (see *Supplementary file 1a*). (**B**) Experimental days (Days 2 and 3) with cycling task (outside MR) and functional magnetic resonance imaging (fMRI) task (inside MR) with the only difference in the drug treatment administered (SAL or NLX). ITI = inter-trial interval; VAS = visual analogue scale.

The online version of this article includes the following figure supplement(s) for figure 1:

**Figure supplement 1.** Effect of exercise intensity, drug treatment, and sex on pressure pain modulation.

**Figure supplement 2.** Distribution of weight-corrected functional threshold power (FTP) for both sexes.

as well as heterogeneous study designs (*Vaegter and Jones, 2020*), further limiting the interpretation of results (*Naugle et al., 2012*; *Wewege and Jones, 2021*).

In this study, we examined exercise-induced pain modulation between different exercise intensities in a healthy sample with a particular emphasis on the opioidergic system. This preregistered study ( drks.de: DRKS00029064) comprised an initial calibration day and two experimental days (*Figure 1*). On the calibration day, individual exercise intensities, pain intensities, as well as objectively assessed fitness levels (functional threshold power; $FTP_{20}$) (*Borszcz et al., 2018*), were determined (*Figure 1A*). On each of the two experimental days, participants cycled during four blocks at high (100% FTP) and low (55% FTP; active recovery) intensity for 10 min per block. After each cycling block, functional magnetic resonance imaging (fMRI) commenced, where participants received nine painful heat stimuli (alternating with nine painful pressure stimuli) with a 15-s plateau each and rated their painfulness on a visual analogue scale (VAS) ranging between the calibrated pain threshold (minimally painful) and pain tolerance (almost unbearably painful). Furthermore, we employed a double-blind cross-over pharmacological challenge across experimental days, where either saline (placebo; SAL) or the μ-opioid antagonist naloxone (NLX) was administered intravenously before the experiment commenced and a constant dose was maintained throughout the experiment (see Methods for details) (*Figure 1B*).

## Results

In our preregistration, we also included pressure pain to investigate exercise-induced pain modulation across different pain modalities. However, pressure pain did not show a significant effect of exercise in any of the analyses performed (*Figure 1—figure supplement 1*); therefore, this report focuses on heat pain. As the inter-trial interval (ITI) between heat pain stimuli needs to be about 30 s due to the stimulus length (*Tran et al., 2010*), the interspersed pressure pain stimuli allowed us to additionally (1) characterise the brain responses to pressure as compared to heat pain and (2) investigate the effect of a μ-opioid antagonist on these responses. However, these results are beyond the scope of this paper and will be reported separately. The final sample included healthy female (*n* = 21) and male (*n* = 18) participants of varying fitness levels (*Figure 1—figure supplement 2*). The equal distribution of menstrual phases (based on self-report and divided into three phases: follicular, ovulatory, and luteal) for the experimental days was estimated using a $\chi^2$-test (*Wilson, 1927*) and showed neither significant differences on each experimental day nor between days (see *Supplementary file 1b*).

### Effective induction of heat pain

As a first step, we verified the successful application of the calibrated heat pain intensities in the placebo (saline; SAL) condition. A significant main effect of stimulus intensity on heat pain ratings was observed ($\beta$ = 1.42, *CI* [1.37,1.48], SE = 0.03, *t*(1358.65) = 51.84, p < 2 × $10^{-16}$; *Figure 2A*, *Supplementary file 1c, d*). Corroborating this, our fMRI data showed blood oxygenation level-dependent (BOLD) activation changes in key brain regions associated with pain processing such as the right anterior Insula (antIns; *Figure 2B*; $MNI_{xyz}$: 36, 6, 14, *T* = 10.35, $p_{corr-WB}$ < 0.001), right dorsal posterior insula (dpIns; *Figure 2C*; $MNI_{xyz}$: 39, –15, 18, *T* = 7.65, $p_{corr-WB}$ < 0.001), and right middle cingulate cortex (MCC; *Figure 2D*; $MNI_{xyz}$: 6, 10, 39, *T* = 7.47, $p_{corr-WB}$ = 0.001). The parameter estimates from the respective peak voxels mirrored the behavioural response pattern (*Figure 2E–G*). Overall, this confirms the successful administration of thermal heat stimuli at the intended intensities. For an uncorrected activation map at $p_{uncorr}$ < 0.001 of the entire brain, see *Figure 2—figure supplement 1*.

### Implementation of exercise intervention

As a next step, we aimed to confirm the successful implementation of our exercise intervention. This intervention comprised previously calibrated high-intensity (HI) and low-intensity (LI) exercise conditions, where the anticipated intensities were derived from the individual FTP value determined on Day 1 (*Figure 1A*). The anticipated power during HI power was set to 100% whereas LI power was set to 55% of FTP. As intended, the maintained power during cycling was significantly higher in the HI exercise condition compared to the LI exercise condition (*t*(37) = 16.09, p < 2.2 × $10^{-16}$, mean power_diff = 60.26 W, *d* = 0.61, *n* = 38). Furthermore, the relative power (as % of FTP) was significantly different between the exercise conditions (*t*(37) = 44.58, p < 2.2 × $10^{-16}$, *d* = 6.46, *n* = 38; *Figure 3A*). Supporting this, the mean absolute heart rate (HR) was significantly higher in the HI exercise condition

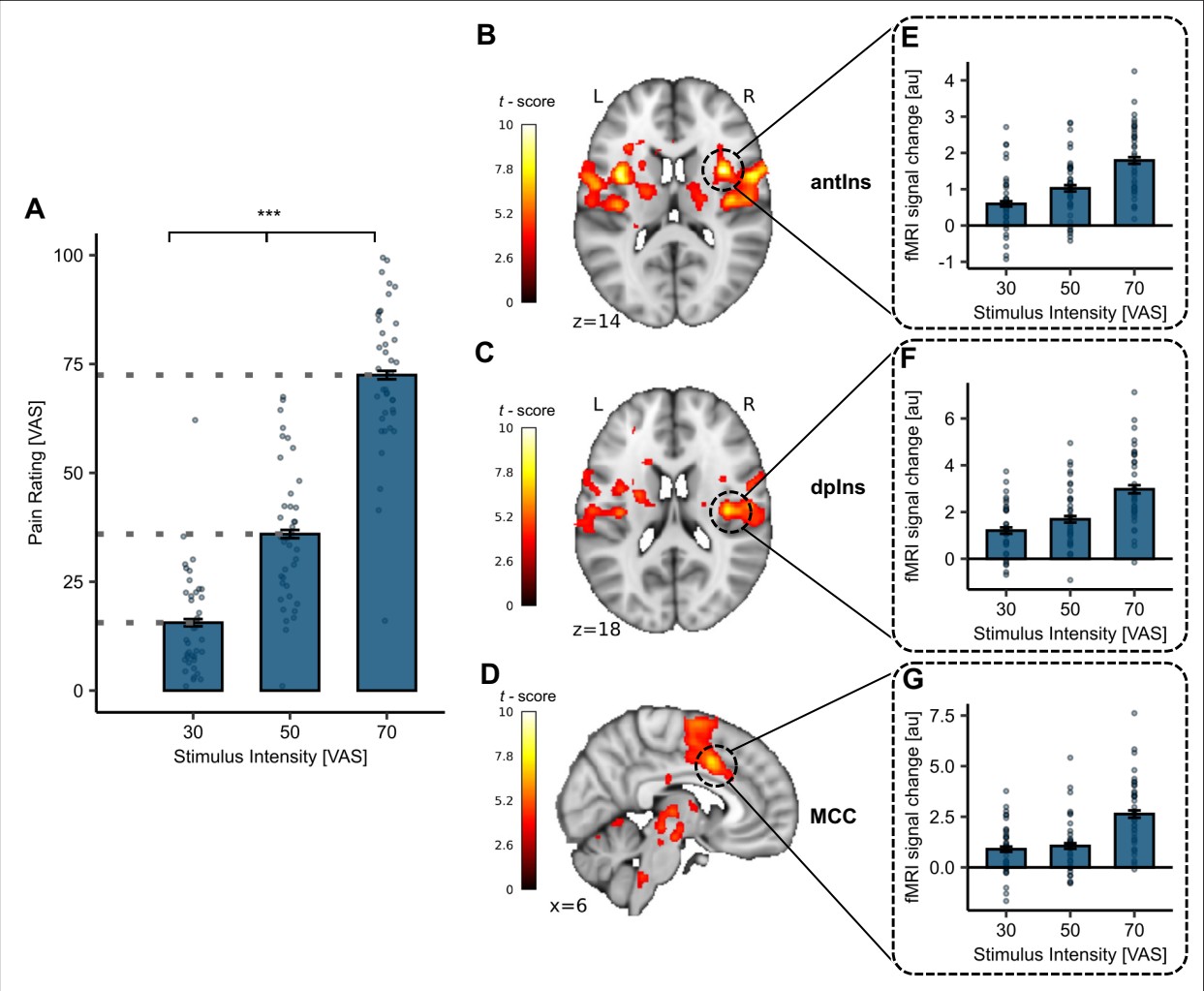

**Figure 2.** Behavioural and functional magnetic resonance imaging (fMRI) results for successful pain induction. (**A**) Heat pain ratings in the SAL condition for all stimulus intensities (visual analogue scale [VAS] 30, 50, 70) showed a significant main effect of stimulus intensity (p < 2 × 10⁻¹⁶). The p-value reflects the significant main effect of stimulus intensity from the linear mixed effect model (LMER). Significant activation for the parametric effect (heat VAS 70 > 50 > 30) in the SAL condition in the (**B**) right antIns (MNI$_{xyz}$: 36, 6, 14; T = 10.35, p$_{corr-WB}$ < 0.001), (**C**) right dpIns (MNI$_{xyz}$: 39, –15, 18, T = 7.65, p$_{corr-WB}$ < 0.001), and (**D**) right middle cingulate cortex (MCC) (MNI$_{xyz}$: 6, 10, 39; T = 7.47, p$_{corr-WB}$ = 0.001). Displayed are the uncorrected activation maps (p$_{uncorr}$ < 0.001) for visualisation purposes. (**E–G**) Bars depict mean parameter estimates from the respective peak voxels for all stimulus intensities across participants, whereas dots display subject-specific mean parameter estimates at the respective stimulus intensity. p-values were calculated using the LMER model for the fixed effect of stimulus intensity. *p < 0.05, **p < 0.01, ***p < 0.001. Error bars depict SEM (N = 39).

The online version of this article includes the following figure supplement(s) for figure 2:

**Figure supplement 1.** Uncorrected activation map for parametric contrast (visual analogue scale [VAS] 70 > 50 > 30) of heat pain in saline (SAL) condition (p < 0.001).

compared to the LI exercise condition (t(33) = 20.48, p < 2.2 × 10⁻¹⁶, mean bpm$_{diff}$ = 26.92, d = 1.85, n = 34, *Figure 3B*). In one participant, the power recording was incomplete (n = 38), and five participants had faulty HR recordings in one session (n = 34) due to an instrumentation error. Nevertheless, the actual power and HR maintained throughout the blocks were also visually monitored. Furthermore, participants rated their perceived level of exertion (RPE) on a BORG scale (*Borg, 1998*) from 6 to 20 following each exercise block. The mean RPE across all HI and LI exercise blocks was also significantly higher after HI compared to LI exercise (t(38) = 19.65, p < 2.2 × 10⁻¹⁶, mean RPE$_{diff}$ = 6.32, d = 3.69, n = 39; *Figure 3C*). Thus, the calibration and implementation of the exercise intensities were successful. To account for a potential expectation about the effect of acute exercise on pain, we have conducted a questionnaire (*Lindheimer et al., 2020*) on the calibration day (Day 1) to capture this.

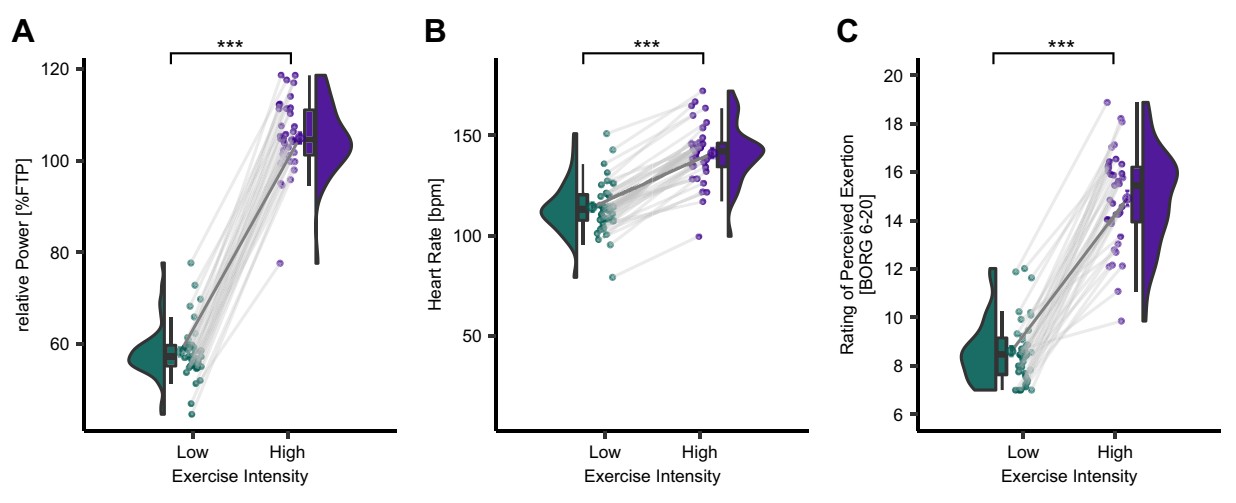

**Figure 3.** Successful implementation of high- (HI) and low-intensity (LI) exercise. (**A**) Relative power (%FTP), (**B**) heart rate in beats per minute (bpm), and (**C**) rating of perceived exertion (RPE; BORG scale) during LI- (green) and HI- (purple) cycling were all significantly different. p-values were calculated using a paired *t*-test (two-tailed, power: *N* = 38, heart rate: *N* = 34, BORG: *N* = 39). *p < 0.05, **p < 0.01, ***p < 0.001.

The online version of this article includes the following figure supplement(s) for figure 3:

**Figure supplement 1.** Expectation of acute exercise on pain.

However, there were no significant expectation effects on different pain domains evident (*Figure 3—figure supplement 1*) suggesting that expectation is unlikely to have influenced the results.

## The effect of drug treatment in response to stimulus intensity

Next, we investigated the effect of our drug treatment with the μ-opioid antagonist naloxone (NLX). Previous studies have shown that NLX can increase perceived pain by blocking the effect of endogenous opioids (*Eippert et al., 2009*; *Schoell et al., 2010*). In line with those findings, we observed a significant interaction of drug and stimulus intensity on heat pain ratings ($\beta$ = 0.10, CI [0.02, 0.18], SE = 0.04, *t*(2755) = 2.46, p = 0.01; *Figure 4A*, *Supplementary file 1e, f*), where, with increasing stimulus intensities, the differences between ratings in the NLX and SAL condition increased ($\beta$ = 2.00, *CI* [0.40, 3.61], SE = 0.81, *t*(77) = 2.46, p = 0.02; *Figure 4B*, *Supplementary file 1g*). We also investigated this interaction of drug and stimulus intensity between sexes and observed this interaction in females ($\beta$ = 0.17, *CI* [0.06, 0.28], SE = 0.06, *t*(1471) = 3.00, p = 0.003; *Figure 4C* and *Supplementary file 1h*) but not in males ($\beta$ = 0.02, *CI* [–0.09, 0.13], SE = 0.06, *t*(1255) = 0.32, p = 0.75; *Figure 4D* and *Supplementary file 1i*). The difference in pain ratings between the drug treatment conditions revealed a significant main effect of stimulus intensity ($\beta$ = 3.41, *CI* [1.27, 5.54], SE = 1.09, *t*(76) = 3.12, p = 0.003) and sex ($\beta$ = 9.75, *CI* [1.23, 18.23], SE = 4.42, *t*(101.60) = 2.21, p = 0.03; *Figure 4E* and *Supplementary file 1j*). In contrast to females, males showed an overall large drug effect at every stimulus intensity. These findings demonstrate a robust drug effect, where pain ratings increased by blocking μ-opioid receptors. The increasing magnitude of the drug effect with increasing stimulus intensities indicates that the opioidergic system might be more extensively employed by more intense noxious stimuli. Since this magnitude differs for lower intensities between males and females, it also indicates a possible sex-dependent effect regarding endogenous opioids.

In the next step, we identified brain regions associated with pain processing and opioidergic pain modulation. We observed a significant activation (contrast interaction of stimulus intensity and drug treatment) in the PAG (MNI$_{xyz}$: –2, –24, –8, T = 5.17, p$_{corr-SVC}$ = 0.01; *Figure 5A*). The extracted parameter estimates from the peak voxel showed that NLX increased activation compared to SAL, especially for the highest stimulus intensity (*Figure 5B, C*). We used post-stimulus averaging (Finite Impulse Response or FIR model) to extract the mean time course for the highest-intensity stimulus (VAS 70). A significant difference in activation was evident 7 s after stimulus onset (BOLD response) for the remainder of the stimulus in the peak voxel in the PAG (*Figure 5D*; for an uncorrected map at p$_{uncorr}$ < 0.001 see *Figure 5—figure supplement 1*). We also investigated the contrast heat NLX > heat SAL

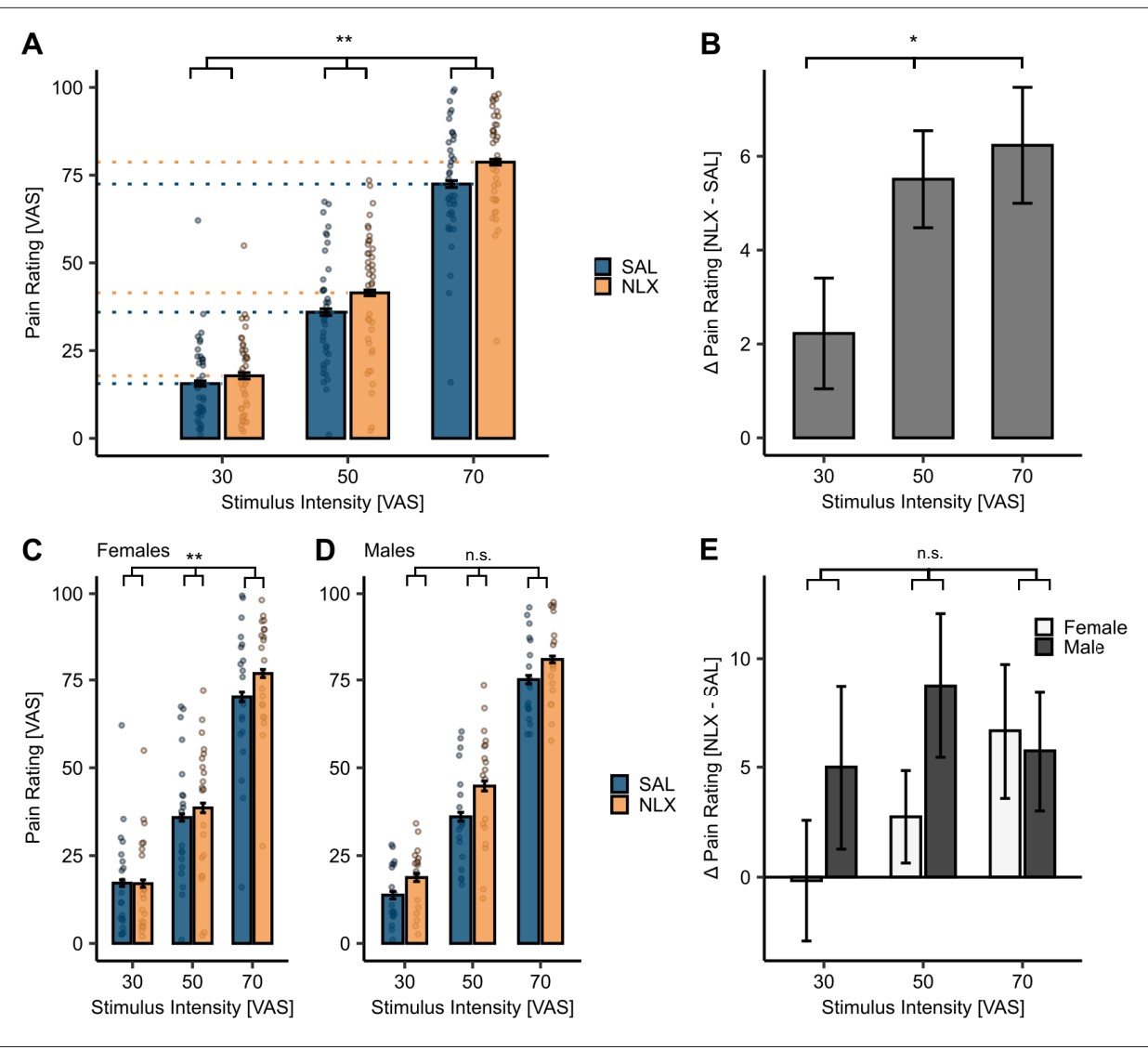

**Figure 4.** Behavioural results for the effect of drug treatment on pain. (**A**) Heat pain ratings revealed a significant interaction between drug and stimulus intensity (p = 0.01). Mean heat pain ratings were significantly higher under naloxone (NLX) treatment (orange) compared to placebo (SAL) (blue) across stimulus intensities. The p-value indicates a significant interaction effect of stimulus intensity and drug. (**B**) Differences in heat pain ratings between NLX and SAL condition (NLX – SAL) at each stimulus intensity revealed a significant main effect of stimulus intensity (p = 0.02). (**C, D**) Heat pain ratings at all stimulus intensities in both drug treatment conditions are significantly higher in NLX (orange) compared to SAL (blue) conditions for both sexes. In females (**C**), a significant interaction of stimulus intensity and drug was evident (p = 0.003) but not in (**D**) males (p = 0.75). (**E**) Differences in heat pain ratings between NLX and SAL condition (NLX – SAL) at each stimulus intensity for females (white) and males (dark grey) revealed a significant main effect of stimulus intensity (p = 0.003) and sex (p = 0.03) with an interaction showing a trend (p = 0.06). p-values were calculated using the LMER model for the interaction of stimulus intensity and drug. *p < 0.05, **p < 0.01, ***p < 0.001. Error bars depict the SEM (N = 39).

where activation in the right antIns showed a trend at whole-brain correction (MNI$_{xyz}$: 46, 8, 6, *T* = 5.70, p$_{corr-WB}$ = 0.08; *Figure 5—figure supplement 2*).

## No difference between HI and LI aerobic exercise on pain modulation

After we established a successful exercise and drug intervention, we investigated the modulating effect of exercise intensity on pain ratings and neuronal responses in the SAL condition. There was no main effect of exercise intensity on pain as there was no hypoalgesic effect evident in the behavioural pain ratings comparing HI to LI exercise in the SAL condition ($\beta$ = 1.19, *CI* [–1.85, 4.22], SE = 1.55, *t*(1354) = 0.77, p = 0.44; *Figure 6A*, blue bars and *Supplementary file 1k*). This contrasts our preregistered hypothesis. Furthermore, there were no significant differences in BOLD activation between

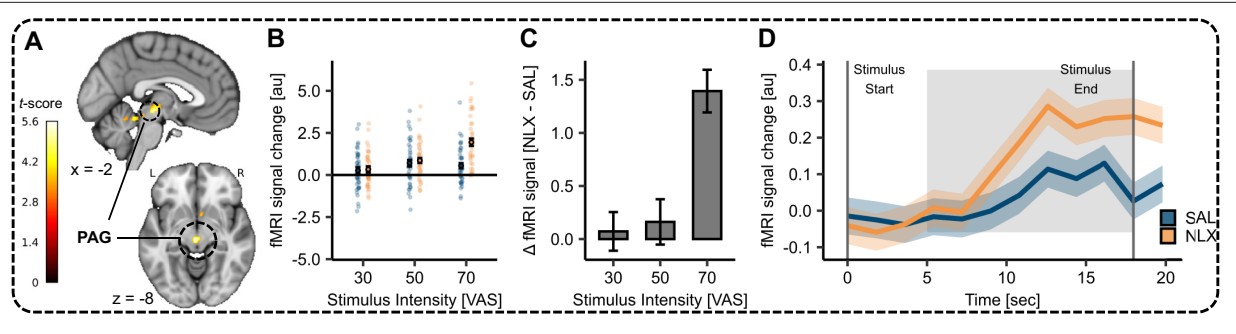

**Figure 5.** Effect of NLX in the periaqueductal grey (PAG). (**A**) Activation for the interaction of stimulus intensity and drug in the PAG (MNI$_{xyz}$: −2, −24, −8, T = 5.17, p$_{corr-SVC}$ = 0.01) superimposed onto an MNI template brain. (**B**) Parameter estimates for SAL (blue) and NLX (orange) conditions for the respective peak voxel in the PAG (MNI$_{xyz}$: −2, −24, −8). (**C**) Difference between parameter estimates of the peak voxel between NLX and SAL condition at each stimulus intensity. (**D**) Time course of blood oxygenation level-dependent (BOLD) responses for SAL (blue) and NLX (orange) during high pain (visual analogue scale [VAS] 70) in this voxel. The shaded areas depict the SEM. The grey solid lines indicate the start and end of the painful stimulus. The shaded grey area displays the approximate time window for BOLD response taking into account a 5-s delay of the haemodynamic response function.

The online version of this article includes the following figure supplement(s) for figure 5:

**Figure supplement 1.** Uncorrected activation map for contrast interaction stimulus intensity and drug treatment (p < 0.001).

**Figure supplement 2.** Cortical drug treatment effect in the anterior insula.

both exercise conditions (contrasts: exercise high > low (SAL) or exercise low > high (SAL)). For the uncorrected activation maps at p$_{uncorr}$ < 0.01 please refer to *Figure 6—figure supplements 1 and 2*. Additionally, we extracted the parameter estimates from the preregistered regions of interest (ROIs) (rostral ventral medulla [RVM], PAG, and frontal midline; *Figure 6B–D*) and estimated the LMER models with exercise intensity on parameter estimates in the SAL condition (*Figure 6E–G*, blue bars). None of these models yielded a significant main effect of exercise intensity (*Supplementary file 1l–n*).

Despite not detecting a hypoalgesic effect of exercise intensity in the SAL condition, we investigated whether naloxone might alter the pain ratings depending on the exercise intensity. We estimated an LMER model with exercise intensity and drug treatment which yielded a significant main effect of drug treatment ($\beta$ = 4.50, CI [1.36, 7.64], SE = 1.60, t(2755) = 2.81, p = 0.005) but no significant interaction effect of exercise intensity and drug treatment ($\beta$ = 0.27, CI [–4.17, 4.71], SE = 2.27, t(2755) = 0.12, p = 0.91; *Figure 6A* and *Supplementary file 1o*). For completeness, we have extended this model to include stimulus intensity, yielding no significant interaction of exercise intensity, drug treatment, and stimulus intensity ($\beta$ = –0.05, CI [–0.20, 0.11], SE = 0.08, t(2751) = –0.56, p = 0.58; *Figure 6—figure supplement 3*). Corroborating this, there were no significant differences in BOLD activation for the contrasts interaction drug treatment and exercise intensity (positive and negative weights). For completeness, we visualised the uncorrected BOLD activation maps at p$_{uncorr}$ < 0.01 in *Figure 6—figure supplements 4 and 5*. We again extracted the parameter estimates from the preregistered ROIs (*Figure 6B–D*) across both treatment conditions and estimated the LMER models with the interaction of exercise intensity and drug treatment on parameter estimates from the respective ROIs (*Figure 6E–G*). None of these models yielded a significant interaction effect of stimulus intensity and drug treatment (*Supplementary file 1p–r*). Since we could not establish a difference in the hypoalgesic effect of HI and LI aerobic exercise in the SAL condition, our preregistered hypothesis of reducing the hypoalgesic effect through naloxone administration behaviourally and neuronally could not be confirmed at either of the exercise intensities.

## Potential association of exercise-induced pain modulation with fitness level in the mFC

Although not preregistered, we additionally explored whether accounting for the different fitness levels of our participants in the statistical model can yield an exercise-induced analgesia effect. As a measure of fitness levels, we utilised the FTP (W/kg), and for ease of interpretation, we further used the difference scores between pain ratings after LI and HI exercise bouts (LI – HI exercise pain ratings). Positive difference scores indicate hypoalgesia (i.e. higher pain ratings after LI exercise)

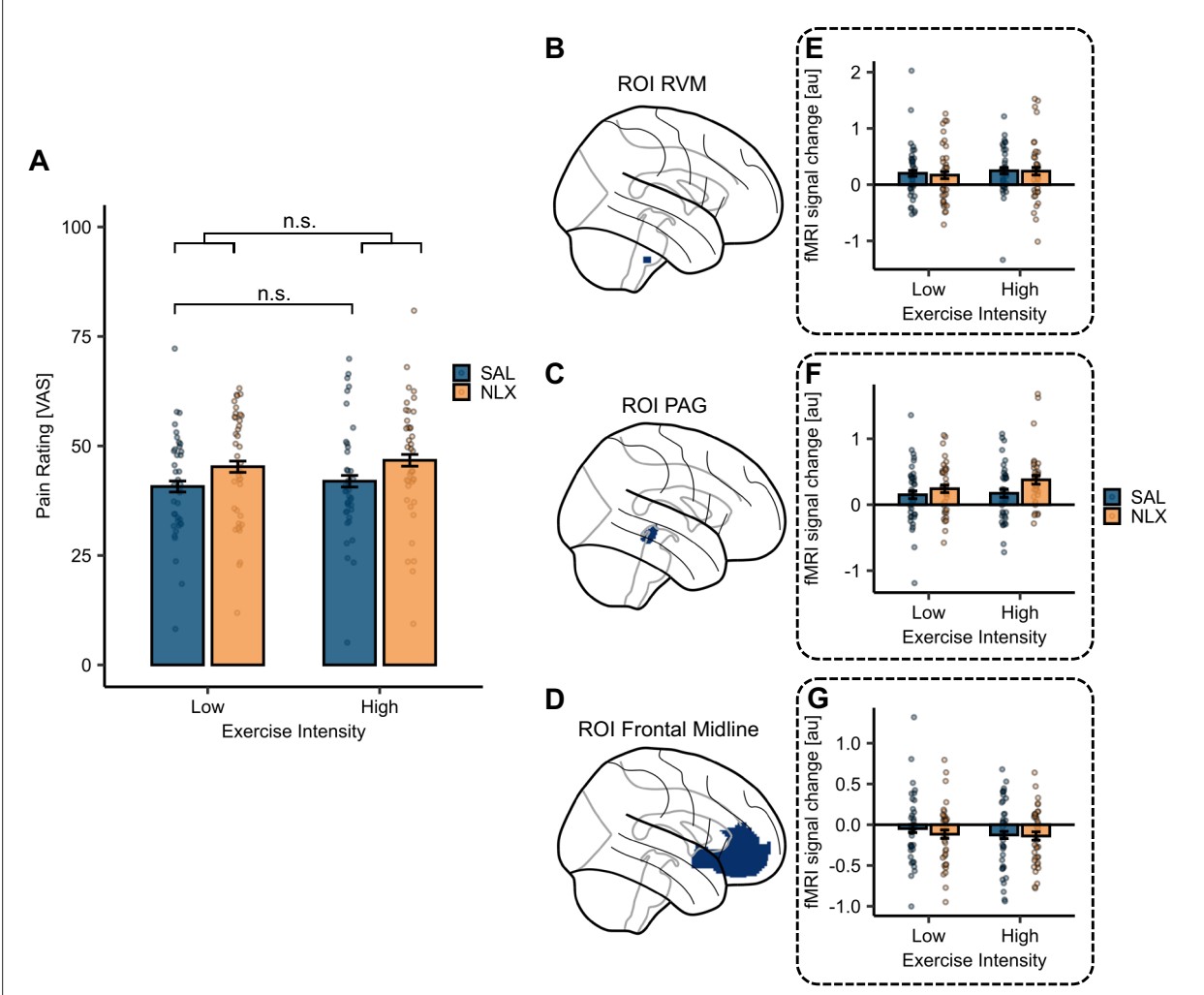

**Figure 6.** Effect of exercise intensity and drug treatment on pain modulation. (**A**) No significant main effect of exercise intensity on pain ratings in the SAL condition (p = 0.44, blue bars). The p-value was calculated using the LMER model with exercise intensity. In a separate LMER model for the interaction of exercise intensity and drug treatment, this interaction effect was not significant (p = 0.91) but a significant main effect of drug treatment (p = 0.005) was evident. Bars depict the average pain ratings in the SAL (blue) and NLX (orange) conditions in both exercise conditions averaged across all stimulus intensities and the dots represent the subject-specific average ratings averaged across all stimulus intensities. Error bars depict the SEM (N = 39). Regions of interest (ROIs) in the (**B**) rostral ventral medulla (RVM), (**C**) periaqueductal grey (PAG), and (**D**) frontal midline (comprised of anterior cingulate cortex [ACC] and ventromedial prefrontal cortex [vmPFC]). (**E–G**) Parameter estimates extracted from both exercise and treatment conditions for the respective ROIs showed no significant effect of exercise intensity in the SAL condition (*Supplementary file 1l–n*) as well as no interaction of stimulus intensity with drug treatment (*Supplementary file 1p–r*). n.s. = not significant, *p < 0.05, **p < 0.01, ***p < 0.001.

The online version of this article includes the following figure supplement(s) for figure 6:

**Figure supplement 1.** Uncorrected activation map for contrast exercise high > low of heat pain in saline (SAL) condition (p < 0.01).

**Figure supplement 2.** Uncorrected activation map for contrast exercise low > high in heat pain in saline (SAL) condition (p < 0.01).

**Figure supplement 3.** Effect of exercise intensity and drug treatment on heat pain ratings at different stimulus intensities (visual analogue scale [VAS] 30, 50, 70).

**Figure supplement 4.** Uncorrected activation map for contrast interaction exercise intensity and drug treatment (pos) of heat pain (p < 0.01).

**Figure supplement 5.** Uncorrected activation map for contrast interaction exercise intensity and drug treatment (neg) of heat pain (p < 0.01).

whereas negative difference scores indicate hyperalgesia (i.e. higher pain ratings after HI exercise). This exploratory analysis showed a significant main effect of fitness level on differences in pain ratings in the SAL condition (*β* = 6.45, *CI* [1.25, 11.65], SE = 2.56, *t*(38) = 2.52, p = 0.02; *Supplementary file 1s*) suggesting increased hypoalgesia with increasing fitness levels after HI compared to LI exercise, pooled across all stimulus intensities (*Figure 7A*). In the brain, the corresponding contrast (exercise

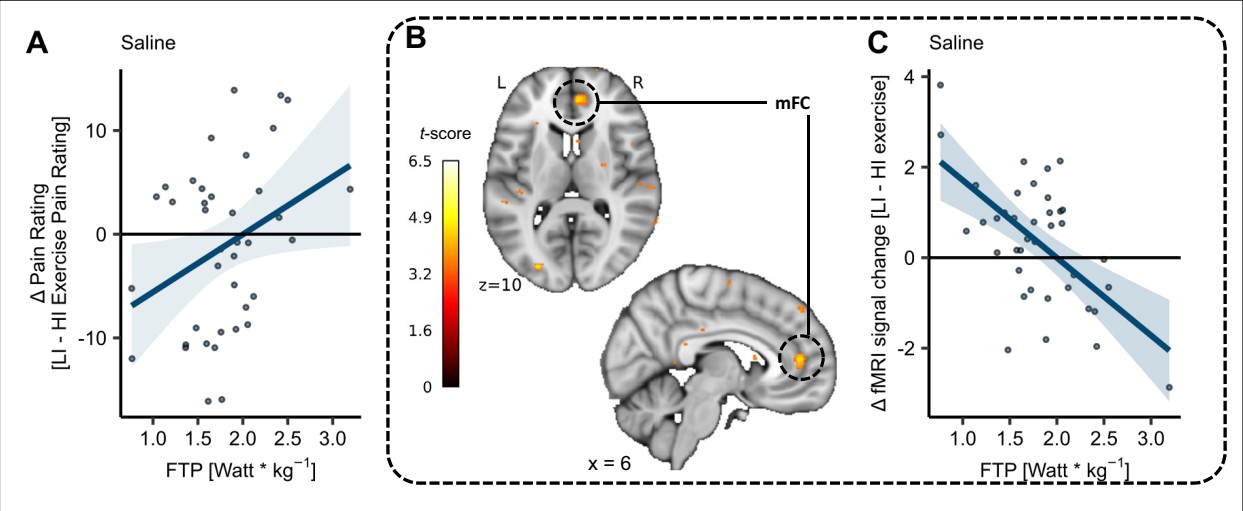

**Figure 7.** Fitness level on difference pain ratings (LI − HI exercise) and medial frontal cortex (mFC) activation. (**A**) Subject-specific differences in heat pain ratings (dots) between low-intensity (LI) and high-intensity (HI) exercise conditions (LI – HI exercise pain ratings) and corresponding regression line pooled across all stimulus intensities in the SAL condition. Fitness level (functional threshold power; FTP) showed a significant positive relation to heat pain ratings with a significant main effect of FTP (p = 0.02) on difference ratings. (**B**) Cortical activation for contrast: exercise high > exercise low with mean-centred covariate FTP (weight-corrected) in the right mFC (MNI$_{xyz}$: 6, 45, 10; $T$ = 4.59, p$_{corr-SVC}$ = 0.05) across all stimulus intensities in the SAL condition superimposed onto an MNI template brain. (**C**) Differences between parameter estimates of LI and HI exercise conditions (LI – HI exercise parameter estimates) from respective peak voxel, plotted for each subject as a function of fitness level (FTP). Regression lines are visualised and shaded areas represent the SEM (*N* = 39).

The online version of this article includes the following figure supplement(s) for figure 7:

**Figure supplement 1.** Effect of exercise intensity and fitness level on absolute pain ratings and medial frontal cortex (mFC) activation.

**Figure supplement 2.** Uncorrected activation map for contrast exercise high > exercise low intensity with covariate functional threshold power (FTP) in saline condition (p < 0.001).

**Figure supplement 3.** Small volume correction mask based on preregistered regions of interest (ROIs).

high > exercise low with the mean-centred covariate FTP (W/kg)) was also investigated in the SAL condition. Here, we observed a significant activation of the mFC (MNI$_{xyz}$: 6, 45, 10; $T$ = 4.59, p$_{corr-SVC}$ = 0.05; *Figure 7B*). The parameter estimates from this voxel revealed that an increase in fitness level was negatively associated with the difference scores between parameter estimates from the drug treatment conditions (*Figure 7C*). These results suggest that the extent and direction of exercise-induced pain modulation potentially depend on the individual fitness levels and might be mediated by the mFC. See *Figure 7—figure supplement 1* for pain ratings and parameter estimates separately for HI and LI exercise and *Figure 7—figure supplement 2* for an uncorrected activation map at p$_{uncorr}$ < 0.001.

### Exploring the effect of sex, fitness level, and endogenous opioids on exercise-induced pain modulation along the mFC

As a second, not preregistered, exploratory analysis, we tested for sex effects in exercise-induced pain modulation. This was motivated by our preceding analyses suggesting that males and females differ in the modulation of pain by NLX administration at different stimulus intensities (*Figure 4E*). We observed a significant interaction of sex, fitness level, and drug ($\beta$ = −13.12, *CI* [−23.69, −2.56], SE = 5.42, *t*(190) = −2.43, p = 0.016, *Figure 8A, B, Supplementary file 1t*). In the SAL condition, males showed larger hypoalgesia with increasing fitness levels (*Figure 8A*, red line) whereas females showed no hypoalgesic response regardless of fitness levels (*Figure 8A*, blue line). In the NLX condition, however, there was no significant association between fitness levels and difference in pain score in males (*Figure 8B*, red line) but a small positive correlation in females (*Figure 8B*, blue line). These results possibly hint at diverging pain modulatory mechanisms in males and females depending on their fitness level and drug treatment. In the next step, we investigated this interaction of fitness level, sex, and drug in the neuroimaging data, formally implemented as a two-sample (male vs. female)

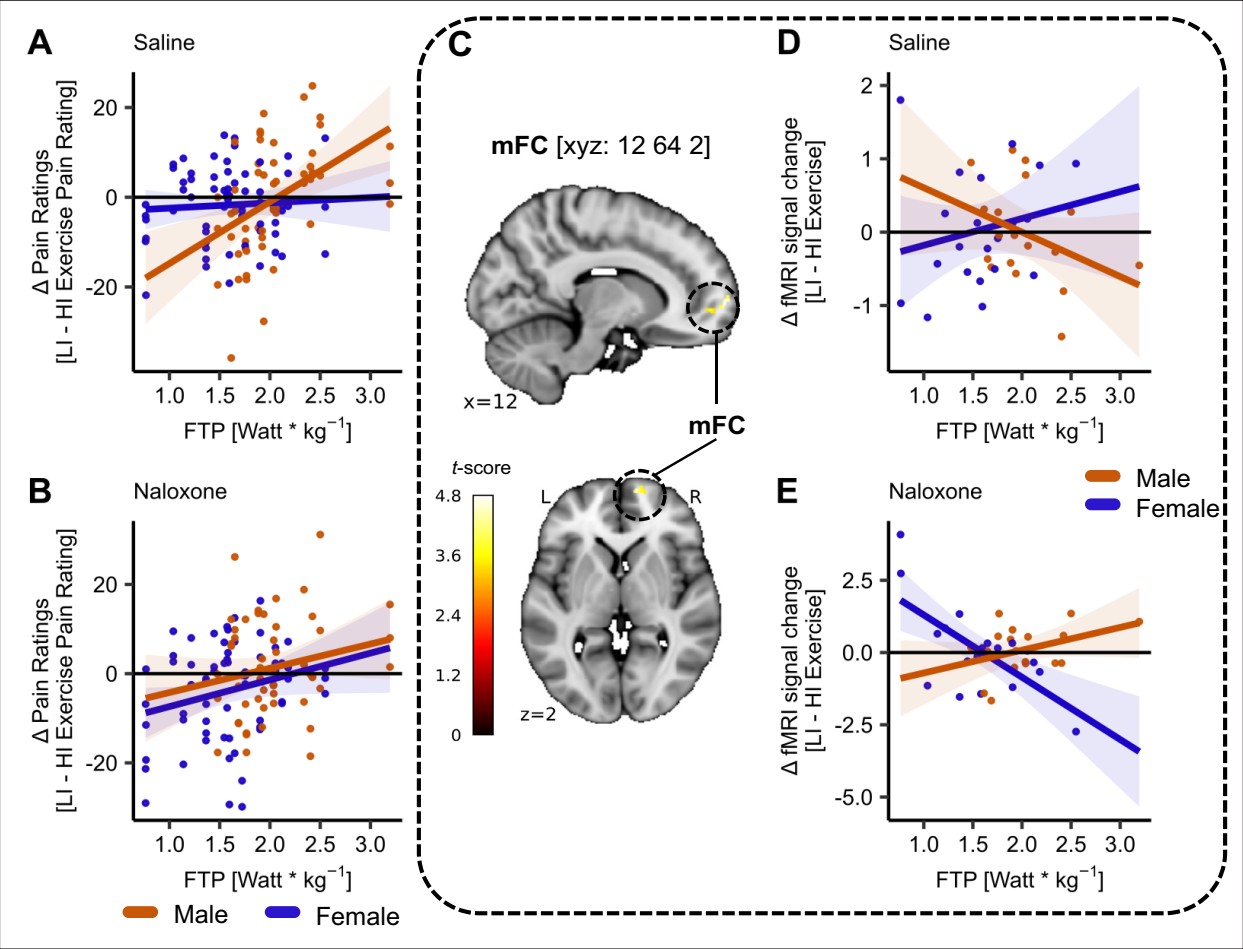

**Figure 8.** Exercise-induced pain modulation potentially depends on sex, fitness level, and drug treatment in medial frontal cortex (mFC). (**A**) Exercise-induced pain modulation in the SAL condition for males (red) and females (blue). Males showed larger hypoalgesic responses with increasing fitness levels as indicated by positive difference ratings in the SAL condition. Females showed no association between fitness levels and difference ratings. (**B**) No exercise-induced pain modulation after blocking μ-opioid receptors with NLX in males (red) and females (blue). (**C**) The activation pattern in the mFC (MNI$_{xyz}$: 12, 64, 2; T = 4.78, p$_{corr-SVC}$ = 0.039) resulting from a two-sample t-test (two-tailed) between males and females for contrast interaction of exercise and drug with covariate functional threshold power (FTP) superimposed onto an MNI template brain. The difference in parameter estimates from the peak voxel in the mFC in (**D**) SAL and (**E**) NLX condition for males (red) and females (blue). Each dot represents the difference in parameter estimates between LI and high-intensity (HI) exelow-intensityrcise conditions (LI – HI exercise) for each subject averaged across all stimulus intensities. Shaded areas represent the SEM (female: N = 21, male: N = 18).

The online version of this article includes the following figure supplement(s) for figure 8:

**Figure supplement 1.** Uncorrected activation map for two-sample t-test between males and females for contrast: interaction exercise and drug with covariate functional threshold power (FTP; p < 0.001).

t-test for the interaction contrast between exercise intensity and drug with the subject covariate of individual FTP. This revealed significant activation in the mFC (MNI$_{xyz}$: 12, 64, 2; T = 4.78, p$_{corr-SVC}$ = 0.039; *Figure 8C*). The parameter estimates from the peak voxel are shown for both drug treatment conditions and both sexes in *Figure 8D, E*. In the SAL condition, males showed a negative association of the difference (EIH) parameter estimates with fitness level (*Figure 8D*, red line), whereas females showed a positive association (*Figure 8D*, blue line). Importantly, this pattern was abolished under NLX (*Figure 8E*; see *Figure 8—figure supplement 1* for an uncorrected activation map at p$_{uncorr}$ < 0.001). Despite being an exploratory analysis, these findings suggest an interaction between sex and fitness level in opioidergic mechanisms that might mediate exercise-induced pain modulation.

## Discussion

This double-blind cross-over study aimed to disentangle the neural and pharmacological mechanisms of exercise-induced pain modulation in healthy individuals of varying fitness levels. There was no main effect of exercise-induced pain modulation after HI exercise compared to LI exercise in the saline (SAL) condition, conflicting our preregistered hypothesis. What is more, naloxone (NLX) administration resulted in increased pain ratings across both exercise conditions but showed no interaction with exercise intensity. These results suggest that there is no difference between HI and LI aerobic exercise on heat pain and, further, no direct involvement of the endogenous opioid system. In an exploratory analysis, we observed a significant interaction involving fitness level, sex, and drug administration where males exhibited greater hypoalgesia, particularly as fitness levels increased under the saline (SAL) condition. This effect was diminished upon administration of naloxone (NLX) and was also apparent in opioidergic regions of the pain modulatory system, namely the mFC.

As expected, we were able to successfully induce heat pain at different stimulus intensities, mirrored by different levels of activation in the right anterior insula (antIns), right dorsal posterior insula (dpIns), and right MCC. As part of a highly distributed pain processing system (*Coghill, 2020*; *Tracey and Mantyh, 2007*), the antIns has been established to encode pain intensity (*Coghill, 2020*) and salience (*Wiech et al., 2010*), whereas the dpIns has been shown to encode stimulus intensity and salience specific to heat pain (*Horing et al., 2019*; *Segerdahl et al., 2015*). The function of the MCC has been more equivocal since some studies attributed a domain-general function of integrating pain, negative affect, and cognitive control (*Shackman et al., 2011*), whereas others argued in favour of a domain-specific representation of painful stimulation (*Kragel et al., 2018*).

### Opioidergic mechanisms in heat pain

Previous research has shown that heat pain is modulated by the descending pain modulatory system including the PAG (*Borras et al., 2004*; *Petrovic et al., 2002*) mainly through opioidergic mechanisms such as placebo analgesia (*Eippert et al., 2009*; *Petrovic et al., 2002*; *Tinnermann et al., 2022*). Accordingly, studies showed VAS ratings of intense heat stimuli to be increased after NLX administration (*Borras et al., 2004*; *Kut et al., 2011*), along with a significant modulation of BOLD responses in the PAG (*Eippert et al., 2009*). In line with this, we could show that NLX significantly increased perceived painfulness mirrored by activation in the PAG (*Corder et al., 2018*). An important aspect of our study is that the magnitude of opioidergic pain modulation increased with pain intensity in females, whereas males showed a constant, but high magnitude of pain modulation across stimulus intensities when NLX was administered. This finding suggests two things: firstly, endogenous pain modulation through μ-opioid receptors increases with increasing pain intensity and, secondly, the magnitude of this modulation differs between sexes. Since our study employed a within-subject cross-over design with an established pharmacological intervention (*Leknes and Atlas, 2020*; *Trøstheim et al., 2023*), we ensured our results were unbiased and highly robust.

### No difference between HI and LI aerobic exercise on pain modulation

We did not detect significant differences in pain ratings and BOLD signal between the exercise conditions (HI and LI) in the SAL condition. Several factors might contribute to this null finding. For one, the exercise intensities employed in this study might not have been suitable to induce pain modulation. Most studies using peak oxygen consumption ($VO_{2max}$) (*Koltyn et al., 1996*; *Micalos and Arendt-Nielsen, 2016*; *Vaegter et al., 2014*) or the maximum heart rate ($HR_{max}$) (*Gomolka et al., 2019*; *Gurevich et al., 1994*; *Jones et al., 2017*; *Jones et al., 2019b*; *Vaegter et al., 2014*) as a measure of exercise intensity were able to induce hypoalgesia with HI exercise ranging between 65% and 75% $VO_{2max}$ and above 70% $HR_{max}$. Previous research has further suggested that HI exercise produces greater hypoalgesia compared to LI exercise (60–70% $HR_{max}$ vs. light activity: *Jones et al., 2019b*; 70% vs. 50% $HR_{max}$: *Naugle et al., 2014*; 75% vs. 50% $VO_{2max}$: *Vaegter et al., 2014*). What is more, previous studies have often employed a resting control condition, which somehow limits the comparability of exercise conditions since a resting baseline does not control for unspecific factors such as cognitive pain modulation (i.e. attentional load or distraction) that is mediated by endogenous opioids (*Brooks et al., 2017*; *Sprenger et al., 2012*) and, thus, potentially exercise. Furthermore, different aerobic exercise durations have been shown to induce hypoalgesia with durations ranging between 8 and 2 min (for review see *Koltyn, 2002*). Previous studies were able to induce hypoalgesia at 10–15 min

of HI aerobic exercise (75% $VO_{2max}$: *Gomolka et al., 2019*; 63% $VO_{2max}$: *Gurevich et al., 1994*; self-paced: *Haier et al., 1981*; 60–70% $HR_{max}$: *Jones et al., 2019b*; 85% $HR_{max}$: *Sternberg et al., 2001*; 75% $VO_{2max}$: *Vaegter et al., 2015*). Thus, we opted for the control condition of LI exercise instead of rest to also add to the knowledge of a dose–response relationship of exercise intensity needed to induce a hypoalgesic effect. We confirmed the intensities of both exercise conditions by physiological (HR) and psychophysiological measures (rating of perceived exertion). Participants indicated significant differences in HR and RPE between both exercise conditions, confirming the actual and perceived distinction between the exercise conditions.

Another factor to consider is that our study utilised the $FTP_{20}$ test as a tool to determine the highest power output a cyclist can maintain in a quasi-steady state (*Allen and Coggan, 2006*). It is frequently used in studies investigating athletes (*Klitzke Borszcz et al., 2020*; *Mackey and Horner, 2021*; *McGRATH et al., 2019*) and shows a high correlation with other measures such as the $FTP_{60}$ test ($r = 0.88$) (*Borszcz et al., 2018*). Compellingly, the $FTP_{20}$ test does not rely on approximations or self-reports but on the actual power maintained (*Allen and Coggan, 2006*). Previous reviews have shown that hypoalgesia following exercise is most reliably induced by HI (i.e. >75% $VO_{2max}$) exercise (*Koltyn, 2002*; *Naugle et al., 2012*; *Wewege and Jones, 2021*). Thus, we decided to employ an HI exercise protocol using the FTP threshold as a reliable measure. Nevertheless, since our study is one of the first studies to employ this $FTP_{20}$ test with a non-athlete, balanced sample of varying fitness levels, one could argue that the $FTP_{20}$ test is not a reliable estimation of the HI exercise condition. Despite research showing that the FTP is a reliable marker of $VO_{2max}$ (*Denham et al., 2020*), we did not directly measure blood lactate levels and $VO_{2max}$ during the exercise and, thus, did not control for the intensity domains (i.e. moderate–heavy–severe). Since the FTP reflects the highest possible intensity by which a steady-state VO2 is maintained (*Sørensen et al., 2019*), the FTP might be a demarcation point between the heavy and severe domains *Jones et al., 2019a*; although this has been criticised (*Wong et al., 2022*). Consequently, some participants might have exercised in the severe domain during the HI exercise and in the moderate domain during the LI exercise. However, as mentioned above, the RPE and relative power (%FTP) still confirmed a successful implementation of the exercise intervention at the anticipated intensities, suggesting the $FTP_{20}$ test to be a reliable estimation of FTP in our sample. Overall, we ensured the highest accuracy in determining the individual exercise intensities at HI and LI.

Furthermore, the composition of the sample, which included non-athletes with varying fitness levels and of both sexes, might have contributed to the missing difference between HI and LI aerobic exercise on pain modulation. In human studies, many findings on exercise-induced pain modulation are derived from studies in athletes or physically active individuals showing increased hypoalgesia following exercise compared to subjects with lower fitness levels (*Crombie et al., 2018*; *Geva and Defrin, 2013*; *Naugle and Riley, 2014*; *Schmitt et al., 2020*). This observation might be related to greater exercise-induced opioid release in structures such as the PAG or the frontal medial cortex (including the ACC) in fitter individuals (*Saanijoki et al., 2022*; *Sluka et al., 2018*). Furthermore, exercise-induced analgesia in rodents and humans has predominantly been studied in all-male samples (*Koltyn et al., 2014*; *Mogil, 2012*; *Niesters et al., 2010*). Of those studies that have considered potential sex differences in exercise-induced pain modulation, some (*Brellenthin et al., 2017*; *Koltyn and t, 2013*; *Naugle et al., 2014*) could not reveal a sex effect, whereas others (*Koltyn, 2000*; *Sternberg et al., 2001*; *Vaegter et al., 2014*) revealed a stronger effect in females. This is in contrast to results in other domains, where opioid-dependent pain relief through conditioned pain modulation (*Bulls et al., 2015*; *Ge et al., 2004*; *Granot et al., 2008*) or placebo analgesia (*Aslaksen et al., 2011*; *Desai et al., 2022*) has been observed to be stronger in males. We deliberately did not employ professional athletes in this study but rather individuals of varying fitness levels of both sexes to make our results more applicable to the general population.

In the context of exercise-induced pain modulation, different types of pain have been investigated (i.e. thermal, pressure, and electrical) as well as different measurements of such (threshold, intensity, and tolerance) (for reviews see *Koltyn, 2000*; *Naugle et al., 2012*; *Vaegter and Jones, 2020*). For aerobic exercise, the highest effect sizes were reported for pressure pain (pain intensity: Cohen's $d = 0.69$) closely followed by heat pain (pain intensity: $d = 0.59$) (*Naugle et al., 2012*). Despite having successfully induced pressure pain using computer-controlled cuff pressure algometry (CPAR), we were unable to detect significant changes in pain ratings for different exercise intensities. However,

most of the studies employing pressure pain used handheld algometry exerting local pressure which can be prone to experimenter bias (*Vaegter and Jones, 2020*) and those results obtained might not apply to cuff pressure pain. Thermal heat pain was the other chosen method to induce pain and, as mentioned above, we were able to successfully induce heat pain at different stimulus intensities, mirrored by different levels of activation in the right antIns, right dpIns, and right MCC. Since our method to induce thermal heat pain corresponds to the method used in previous studies, the results of this pain modality might be more comparable to those of existing studies.

## Exploring the effect of fitness level and sex on exercise-induced pain modulation

In the exploratory analyses, we identified that pain decreases following HI compared to LI exercise were associated with higher individual fitness levels in the SAL condition. In agreement with previous studies (*Geva and Defrin, 2013*; *Saanijoki et al., 2022*; *Schmitt et al., 2020*; *Tesarz et al., 2012*), this suggests a dependence of exercise-induced pain modulation on individual fitness. Recent studies indirectly support this notion and observed higher self-reported physical activity levels to be associated with reduced temporal summation in heat pain (*Naugle and Riley, 2014*) and increased heat pain thresholds to be associated with individual fitness levels in males (*Schmitt et al., 2020*). In a larger cohort, an association between physical activity levels and pressure pain tolerance (*Årnes et al., 2023*) as well as cold pressor pain tolerance (*Årnes et al., 2021*) was observed. Along with our behavioural findings, we identified increased activation in the mFC after HI exercise to be associated with higher fitness levels in the SAL condition. Previous research has reported increased activation in the rACC to be associated with opioid analgesia (*Petrovic et al., 2002*; *Tinnermann et al., 2022*), thus implying opioidergic involvement of this structure in mediating exercise-induced pain modulation (*Petrovic et al., 2002*). Adding to this, the pregenual ACC showed decreased activation in response to painful heat following a 2-hr run compared to before the run in trained male elite runners (*Scheef et al., 2012*). Furthermore, larger decreases in μ-opioid receptor binding in regions including the ACC, bilateral insula, and OFC in healthy males have been observed to be associated with higher self-reported physical fitness levels (*Saanijoki et al., 2022*). This finding could support observations showing that repeated exercise 'trains' the mFC, which has a high μ-opioid receptor density (*Corder et al., 2018*) and thus enables exercise-induced pain modulation even in the context of brief bouts of HI exercise.

Furthermore, we explored the interaction between sex, fitness level, and drug, where males exhibited larger hypoalgesia following HI exercise with increasing fitness levels, which was attenuated by NLX. This pattern was not evident in females. Our brain data also showed this three-way interaction in the mFC. More precisely, in the SAL condition, males showed higher activation after HI exercise compared to LI exercise in association with higher fitness levels, and this pattern was reversed in the NLX condition. Females showed the reversed pattern to males in both SAL and NLX conditions. Since rodent and early human research have also predominantly employed male samples to avoid hormonal-related fluctuations (*Mogil, 2012*; *Mogil and Chanda, 2005*), this might have led to an over-generalisation of opioidergic effects (*Koltyn et al., 2014*; *Mogil, 2012*) whilst neglecting sex differences in opioid analgesia (*Bodnar and Kest, 2010*; *Niesters et al., 2010*). Hence, it comes as no surprise that findings in humans are heterogeneous (*Koltyn, 2000*; *Wewege and Jones, 2021*) with some studies showing no difference in exercise-induced pain modulation between males and females (*Koltyn et al., 2014*; *Koltyn and t, 2013*; *Smith, 2004*) whereas other studies showed hypoalgesia from exercise to be greater in women (*Vaegter et al., 2014*) but most research on exercise-induced pain modulation has been conducted in men. The few existing imaging studies show that frontal midline structures (ACC and OFC) with high opioid receptor concentration are involved in exercise-induced (pain) modulation (*Boecker et al., 2008*; *Saanijoki et al., 2022*; *Scheef et al., 2012*). A recent meta-analysis has shown that increases in the mFC (including the vmPFC) activity along with reduced activation in regions associated with noxious stimuli (i.e. ACC) could be linked to an expectation of reduced pain, in this case, after HI exercise (*Atlas and Wager, 2014*). When comparing athletes and non-athletes, one study identified an interaction of sex and athletic status where female athletes displayed significantly lower cold pressor ratings than female non-athletes, but there was no difference in men (*Smith, 2004*). These exploratory findings suggest sex- and fitness level-dependent effects in exercise-induced pain modulation but do not provide a definite answer. Future research

should further investigate the role of sex and fitness levels concerning exercise-induced hypoalgesia, specifically investigating females with higher fitness levels.

### Limitations

One limitation that should be considered when interpreting the results is the influence of other mediating factors in exercise-induced pain modulation such as endocannabinoids (eCB; *Crombie et al., 2018*; *Siebers et al., 2021*). Previous animal studies have established their crucial influence in modulating mood and promoting anxiolysis following exercise (*Fuss et al., 2015*). In humans, however, the interpretability of results remains limited since only peripheral eCB can be measured and central eCB blockage is difficult.

In conclusion, our study showed no difference in pain modulatory effects between HI and LI aerobic exercise, neither behaviourally nor in brain responses. However, explorative analyses suggested that there is, in fact, an interaction between sex and fitness level in mediating exercise-induced pain modulation through opioidergic mechanisms in the mFC. To thoroughly investigate this, future research should specifically address the link between endogenous opioids, fitness levels, and sex-dependent differences in exercise-induced pain modulation.

## Materials and methods

The study was preregistered with the WHO-accredited *Deutsches Register für Klinische Studien* (DRKS; Study ID: DRKS00029064; German Registry for clinical studies) before data acquisition.

### Experimental design

This randomised control fMRI study employed a pharmacological intervention with the μ-opioid antagonist NLX in a within-subject design and was divided into a calibration and two experimental days (*Figure 1*). On the experimental days, participants cycled outside the MR for 10 min at HI or LI on a stationary bike per block. Afterwards, they rated the perceived painfulness of heat and pressure stimuli inside the MR. This procedure was repeated for four blocks per experimental day. The objective of this study was to identify the behavioural, pharmacological, and neuronal underpinnings of exercise-induced pain modulation.

### Participants

Forty-eight healthy, right-handed participants were invited to take part in the study. In the framework of the within-subject design, all participants underwent all conditions which included an exercise factor (high vs. low exercise intensity) and a pharmacological intervention (NLX vs. SAL). All participants received the pharmacological intervention in a randomised double-blind fashion on separate study days, where the μ-opioid antagonist NLX or placebo SAL was administered i.v. Our final sample size was determined based on previous behavioural studies investigating exercise-induced pain modulation after aerobic exercise in a pre–post measurement design across different pain modalities ($d = 0.59$ for heat pain intensity, $d = 0.69$ for pressure pain intensity) (*Naugle et al., 2012*). According to this, a sample size of $N = 39$ was sufficient to reveal results at effect size $f = 0.3$, (via G*Power 3.1, $\alpha = 0.05$, $1 - \alpha = 0.95$, one group, two measurements with correlation $r = 0.5$). Based on previous fMRI studies, it can also be assumed that a sample size of $N = 30$ would be sufficient to detect changes in the brain (*Mumford, 2012*). In this study, the term 'sex' refers to 'sex assigned at birth' as indicated by self-reports from the participants. Multiple options ('male', 'female', 'other', and free text field) were provided. All participants reported having been assigned the 'female' or 'male' sex at birth.

### Exclusion of participants

Upon entering the study, participants were screened for anxiety (State and Trait questionnaires of the German short version of the State-Trait Anxiety Inventory; STAI) (*Laux et al., 1981*), depression (Beck Depression Inventory 2; BDI-2) (*Beck et al., 2011*) as well as potential MR contraindications. Furthermore, participants' body mass index (BMI) was required to range between 18 and 30. On the first day of the study, an electrocardiogram (cardiofax, Electrocardiograph-1250, Nihon Koden) was recorded to prove no cardiovascular irregularities. The age of participants was between 18 and 45 years. Eight participants were excluded from the study for the following reasons. Two participants withdrew from

the study after the first day due to circumstantial reasons (moving, no appointment found). Two participants were excluded after Day 1 due to an unreliable thermal calibration and a BMI score below 18, respectively. Two participants did not wish to continue the study after Day 2 due to personal reasons. Two participants were excluded after Day 2 due to a syncope when administering the drug (in both cases SAL has been administered). Upon data screening, one subject was excluded due to excessive movement (>0.6 mm difference between volumes within runs in more than 5% of volumes in all runs) in the MR scanner. A total of 39 participants (age: $M$ = 26.03, SD = 4.8, 21 female) were included in the final sample (*Supplementary file 1u*).

## Ethics approval

The study was approved by the Ethics committee of the medical board in Hamburg (Aerztekammer) and conducted following the Declaration of Helsinki (*World Medical Association, 2013*). All subjects provided informed consent upon entering the study after having been informed about the study procedures including thermal and pressure stimulation, the MR procedure, the double-blind nature of the pharmacological intervention, and potential adverse effects of NLX by the study investigator and study physician. Upon completing or leaving the study, participants were informed of the order of pharmacological treatment received by the unblinded study physician.

## Thermal and pressure stimuli

Thermal stimulation was applied using a thermode (TSA-2, Medoc, Israel) attached to the left lower arm where four 2.5 × 2.5 cm squares were drawn below each other and numbered (*Figure 1A*). For the calibration, the thermode head was positioned in the second square. Each thermal stimulus lasted up to 17 s in total with a plateau of 15 s and a ramp speed of 13°C/s (ramp-up, ramp-down). For the pressure calibration, a computer-controlled CPAR (NociTech, Denmark, and Aalborg University, Denmark) consisting of a compressor tube and 13 cm wide tourniquet cuff (VBM medical, Sulz, Germany, 61 cm length) was used. The cuff was mounted to the left upper arm with a 3-cm distance to the cubital fossa to exert pressure on the upper arm tissue (*Graven-Nielsen et al., 2015*). Additionally, a protective tubular elastic dressing (Tricofix, D/5, 6 cm × 20 cm) was positioned underneath the cuff to protect the bare skin from potential harm. The pressure was applied by inflating the chambers of the cuff with a maximal pressure limit of 100 kilopascal (kPa) and a rise speed dependent on the target pressure. The stimulus length of the pressure stimuli was also up to 17 s (ramp-up, 15-s plateau, ramp-down).

## Calibration procedure (Day 1)

The first day of the study took place outside the MR scanner and served as a calibration day (*Figure 1A*). Blood pressure, baseline HR, and oxygen saturation (SPO$_2$) were measured using a blood pressure cuff (boso medicus uno, bosch + sohn) and a pulse oximeter (pulox Pusloximeter, Novidion GmbH), respectively. The thermal and pressure stimuli were calibrated individually but with an identical protocol in a pseudo-randomised order across participants. During calibration, participants remained in a supine position with the blinds down to emulate the inside of the MR scanner bore. As an initial step, participants received two low stimuli (10/20 kPa or 41/42°C) to get acquainted with the stimuli. Following this, the pain threshold was estimated by applying six adaptive stimuli which were rated on a binary scale (painful vs. not painful). The resulting pain threshold was used as a basis for the following linear regression algorithm aiming to collect ratings across different intensities on a VAS (10, 30, 50, 70, 90 VAS). After each stimulus, participants had to rate its painfulness (How painful was the last stimulus?) on a VAS ranging from 0 (minimally painful) to 100 (almost unbearably painful) using the left and right key of a pointer as a button box (Logitech) with their right hand. The calibration was finished once sufficient rating coverage for all target intensities was reached. The thermal and pressure calibration aimed to identify super threshold intensities (°C and kPa) corresponding to 30, 50, and 70 on the VAS scale using this linear regression algorithm yielding comparable intensities across subjects.

## FTP test (Day 1)

As a final step of the calibration day, an FTP test was conducted on a stationary cycle ergometer (KICKR Bike, Wahoo, Atlanta, United States). The FTP test was first described by *Allen and Coggan, 2006* and determines the maximum average power output that can be maintained for 1 hr which serves as an approximation of the anaerobic threshold (*Allen and Coggan, 2012*). The FTP$_{20}$ test is

derived from the original FTP test but estimates 95% of FTP based on 20 min of steady cycling at maximum possible power output. The test is divided into six stages lasting 1 hr in total (*Figure 1A* and *Supplementary file 1a*). Within this test protocol, the average power output measured for the last 20-min interval is taken as 95% of the actual FTP (*Allen and Coggan, 2012*). The FTP$_{20}$ test allows the identification of individually calibrated power zones. In this present study, we used HI exercise (zone 4: threshold, 91–106% of FTP) and LI exercise (zone 1: active recovery, ~55% of FTP) as intensities. During the FTP$_{20}$ test, participants were free to adjust the resistance of the ergometer whilst maintaining a constant cadence. Power output in Watts, HR (HR belt; Garmin HRM-PRO PLUS) in beats per minute (bpm), and cadence (in RPM) were monitored throughout the FTP$_{20}$ test. The current power and HR, as well as the remaining time of each section, were displayed on a 50-by-30 cm screen in front of the participants to allow for self-monitoring. Participants were asked to refrain from any exhausting physical activities the day of and before the calibration day. The FTP value from the FTP$_{20}$ test was corrected for participants' weight (FTP/weight) and served as an indicator of individual fitness level in the analyses. The distribution of the weight-corrected FTP for males and females is visualised in *Figure 1—figure supplement 2*. There was no significant association between heat pain thresholds and weight-corrected FTP ($r = –0.23$, $p = 0.16$). On the bike, the knee angle was adjusted to a 5-degree bend when the pedal stroke was at the bottom (*Vaegter et al., 2019*). The knee was also set to be over the pedal axle when the cranks were parallel to the ground. The upper body was adjusted to a forward lean with a slight bend of the elbows to maintain a neutral spine position with a natural lumbar flexion.

## Experimental paradigm (Days 2 and 3)

The two experimental days were at least 3 days apart (median = 7 days, mean = 12.35 days, SD = 14.5 days) and identical with the only difference being the pharmacological treatment (NLX vs. SAL) received (*Figure 1B*). Each visit took place in the morning with a starting time of 8:15 a.m. and varied by a maximum of 4 hr between participants to account for potential circadian influences. Blood pressure and SPO$_2$ were measured before each cycling block and monitored throughout the investigation. On each experimental day, a urine sample was tested for amphetamines, opiates, marijuana, methamphetamines, morphine (Surestep, Multidrug Test, Innovacon Inc, San Diego, USA), and potential pregnancies (only female participants; hCG Ultra Test, serum/urine 10 mIU, Mexacare GmbH, Germany). Each experimental day consisted of four blocks. Each block included a 10-min cycling block at a HI (100% FTP) or LI (55% FTP) in a pseudo-randomised order across participants (i.e. participants completed two cycling blocks at HI and two at LI exercise). The order of cycling blocks was kept constant across experimental days (Days 2 and 3). Each cycling block was immediately followed by an MR scan including the pain stimulation and ratings that lasted approximately 20 min. After this, participants' HR and SPO$_2$ were measured outside the MR (approximately 5–10 min) before the next block commenced, starting again with 10-min cycling. On the second experimental day (Day 3), an anatomical scan (T1) was acquired alongside conducting a re-calibration of thermal and pressure stimuli. The re-calibration was based on the previously calibrated intensities from the calibration day (Day 1) and was adjusted based on the ratings provided in the scanner corresponding to 30, 50, and 70 VAS. This procedure served to account for potential differences in perception due to the different environment of the MR scanner but also to acquaint participants with the procedure inside the MR to minimise the time of transition between the cycle ergometer and MR scanner.

## Drug administration

The procedure of drug administration and dosage was based on a previous study using NLX (*Eippert et al., 2009*) as well as a web-based application for detailed planning of μ-opioid antagonist administration (*Trøstheim et al., 2023*). Participants were positioned in a supine position before administering a bolus dose of NLX (0.15 mg/kg; Naloxon-ratiopharm 0.4 mg/ml injection solution, Ratiopharm, Ulm, Germany) or SAL (Isotone Kochsalz-Lösung 0.9% Braun) via peripheral venous access in the right antecubital vein. To ensure correct administration, two participants received intravenous access on the back of their right hand upon giving verbal consent and ensuring no pain was caused by the location of the intravenous line. Shortly after administering the bolus dose, the intravenous infusion dose of NLX (0.2 mg/kg/hr diluted in SAL) or SAL was started using an infusion pump (Perfusor Space, Braun, Munich, Germany). After ensuring that no pain was caused by the intravenous line and that

participants felt comfortable, the first cycling block commenced. The first 10-min cycling block also served to ensure that NLX reached a peak level in the blood plasma (*Trøstheim et al., 2023*) before positioning participants in the MR scanner. The infusion dose was constantly running throughout the whole duration of the study, including cycling blocks outside as well as pain stimulation inside the MR, providing a constant supply of NLX or SAL. Considering the relatively short half-life of NLX (~70 min in blood plasma; Summary of Product Characteristics, Ratiopharm), this procedure ensures a steady-state concentration of NLX for the duration of the study (*Trøstheim et al., 2023*) with an effective central opioid block throughout the study (*Eippert et al., 2009*; *Trøstheim et al., 2023*). The experimenter (J.N.) and research assistants who interacted with the participants were blinded to the pharmacological intervention and remained blinded throughout the study. The research assistant administering the drug treatment was also blind to the treatment. Only the study physician (T.F.), who did not interact with the participants on the experimental days of the study, was unblinded as to what drug was administered. Unblinding of the participants took place after the experiment by the study physician without anyone else present to prevent expectation induction in the experimenter and student assistants. Before unblinding the participants, they were asked what treatment they received on which experimental day (incorrect: $n = 4$, unsure/identical: $n = 19$, correct: $n = 16$).

## Questionnaires

On the calibration day (Day 1) participants completed the short form of the Profile of Mood States (POMS) (*Curran et al., 1995*), STAI (*Laux et al., 1981*), BDI-2 (*Beck et al., 2011*), and a questionnaire on the expectation of exercise in psychological and physiological domains. Furthermore, participants' sleep quality, food intake, menstrual cycle phase, smoking, alcohol, and caffeine consumption on the study day and before the study days were assessed. The menstrual cycle phase was assessed by self-report on each experimental day where female participants indicated which phase applied to them (three phases: follicular, ovulatory, and luteal or hormonal contraceptives). Four female participants indicated they were on hormonal contraceptives. On both experimental days (Days 2 and 3), participants filled out the POMS before and after the study to monitor potential mood changes (results are reported in *Supplementary file 1v*). Furthermore, participants were asked to fill out questionnaires including BDI, STAI, and a questionnaire concerning the side effects of NLX/SAL after both experimental days (Days 2 and 3; results are reported in *Supplementary file 1w*). On the last experimental day (Day 3), participants had to indicate which pharmacological treatment they suspected to have received on which experimental day to capture potential expectations about the pharmacological treatment.

## Data acquisition

### Behavioural data acquisition

During the cycling blocks, HR, power output (Watts), and cadence (RPM) were recorded using MATLAB 2021b. For each cycling block, the required power as well as the actual power maintained by the participant was displayed on a screen to allow for self-monitoring. During cycling, participants' HR and $SPO_2$ were constantly monitored and recorded. In one participant, the recording of the power output failed, and in five participants, HR recordings in one session failed due to equipment issues. However, power output and HR maintained throughout cycling were visible and accessible at all times for the participant. Upon completion of the 10-min cycling, participants were asked to rate their perceived exertion (RPE) on the BORG scale (no exertion (6) – maximal exertion (20)) (*Borg, 1998*) as well as to indicate their current mood (sad (0) – happy (10)). The transfer to the MR scanner commenced as quickly as possible (time: mean = 5 min, SD = 1 min) whilst maintaining the participants' safety at all times. Inside the MR scanner, all stimulus presentation was realised using MATLAB R2016b and Psychophysics Toolbox (Version 3.0.19). Physiological data, including respiration and HR, was recorded at 1000 Hz with the Expression System (In Vivo, Gainesville, USA) using the spike2 software and a CED1401 system (Cambridge Electronic Design). A fixation cross and VAS scale were displayed on a screen inside the scanner bore. Thermal stimuli were applied using a TSA-2 Thermode with a 2.5 × 2.5 cm probe attached to the left volar forearm. Pressure stimuli were applied using a pressure cuff attached to the left upper arm and a CPAR outside the MR. The infusion pump was positioned outside the scanner with a 10-m tube reaching into the scanner room to maintain a constant dose of NLX or SAL. After two pre-exposure stimuli of each modality, thermal and pressure stimuli were applied in an

alternating fashion and at different calibrated intensities (randomised at 30, 50, 70 VAS). Each stimulus lasted 17 s in total (ramp-up, plateau 15 s, ramp-down) indicated by a visual cue (red fixation cross). Following each stimulus, the painfulness was rated on a VAS from 'minimally painful' (converted to 0 for analyses) to 'almost unbearably painful' (converted to 100 for analyses) by using a button box. The rating took place within 7 s after which an ITI of 7 s (jittered by 2 s) was introduced before commencing with the next trial. One MRI block consisted of nine thermal and nine pressure stimuli at three intensity levels (30, 50, and 70 VAS). In total, participants spent 15 min per block inside the MR scanner.

## MRI data acquisition

fMRI data was acquired using a 3 Tesla Siemens system (Magnetom PRISMA; Siemens Healthcare, Erlangen, Germany) with a 64-channel head coil. Sixty slices were acquired (2 mm slice thickness) using T2*-weighted echo-planar imaging (EPI) with multiband factor 2 (repetition time (TR): 1.8 s; echo time (TE): 26 ms; flip angle: 70°). The field of view was 240 mm and positioned to include the upper part of the medulla and the brainstem. The voxel size was thus $2 \times 2 \times 2$ mm$^3$. Four volumes at the beginning of each run were discarded. Furthermore, T1-weighted images were acquired for each subject using an MPRAGE sequence on the first experimental day (Day 2). Due to the nature of the study where participants cycled outside the MR scanner and had to be repositioned for each run, shimming and auto-align took place at the beginning of each run and before acquiring EPI images.

## Statistical analyses

### Behavioural statistical analyses

Behavioural statistical analyses were performed using MATLAB 2021b and RStudio (Version 2021.09.1). We used the *lmer* function from the *lme4* package (Version 1.1-35.1) (*Bates et al., 2014*) and the *lm* function from the *stats* package (Version 3.6.2) (*R Development Core Team, 2021*) to conduct linear mixed effect (LMER) and linear models (LM) in R, respectively (*Hox et al., 2010*; *Raudenbush and Bryk, 2001*). For behavioural analyses, pain ratings of the VAS with the endpoints 'minimally painful' and 'almost unbearably painful' were converted to a numerical scale with the endpoints 0 and 100, respectively. In every LMER model, heat pain ratings or differential heat pain ratings (LI–HI pain ratings) served as the dependent variable, the subject as well as the number of pain ratings (1–9 per block) were included as random effects, whereas the order of drug treatment administration (naloxone or saline first) was included as a fixed effect. In the models that included pain ratings from all stimulus intensities (VAS 30, 50, 70), the stimulus intensity was also included as a fixed effect to account for the variance. Post hoc paired samples *t*-tests (two-sided) were calculated on the LMER models using the R package *emmeans* (Version 1.10.0). The Tukey method was used for adjusting the resulting p-values as implemented in the *emmeans* package. In the instances where the LM was used, heat pain ratings or differential heat pain ratings (LI–HI pain ratings) served as the dependent variable, and order of drug administration was included as an additional independent variable.

To verify the successful application of heat pain, the respective pain ratings in the saline condition only were used as the dependent variable in the LMER model and stimulus intensity (VAS 30, 50, 70) served as a fixed effect. The successful exercise intervention was measured by the average power output, HR, and rating of perceived exertion. The power output (Watt) and HR (bpm) were measured throughout the cycling blocks. The ratings of perceived exertion were measured after each cycling block and a mean RPE value was calculated across the low and high exercise conditions. Paired *t*-tests (two-sided) were conducted for the mean power output, HR, and RPE between the low and high exercise intensity conditions. Cohen's *d* has been calculated as effect size for within-subject samples. To verify the successful drug administration, we conducted LMER models for heat pain ratings with drug and stimulus intensity as well as their interaction as fixed effects. The difference between pain ratings between both drug treatment conditions has been calculated and an LMER model with stimulus intensity as a fixed effect has been calculated. Concerning the potentially diverging effect of drug treatment depending on sex, we conducted an LMER model with drug and stimulus intensity on heat pain ratings for males and females separately. Again, the difference between pain ratings between both drug treatment conditions has been calculated and an LMER model with stimulus intensity as a fixed effect has been calculated for both sexes.

To test our hypothesis of exercise-induced pain modulation, an LMER model with the fixed effects of exercise intensity on heat pain ratings has been calculated. To capture the effect more reliably, the

heat pain ratings after HI exercise were subtracted from heat pain ratings after LI exercise. Positive difference scores would indicate hypoalgesia, whereas a negative score would indicate hyperalgesia following HI exercise. In an LM, the fitness level (weight-corrected FTP) served as an independent variable, and the differential heat pain ratings as the dependent variable. To model the interaction of fitness level, sex, and drug, the differential pain ratings served as the dependent variable for the LMER model, the fitness levels (weight-corrected FTP), drug, and sex, as well as the three-way interaction served as fixed effects.

The mood ratings provided on the POMS on each experimental day pre- and post-measurements were summarised in four domains (dejection, fatigue, discontent, drive) and analysed using Wilcoxon signed rank test (*Wilcoxon, 1945*). There were no significant differences in the post-treatment measurements between the drug treatment conditions (*Supplementary file 1v*). Furthermore, potential differences in side effects were analysed between the drug treatment conditions using the Wilcoxon signed rank test (*Wilcoxon, 1945*) and showed no significant differences between the drug treatment conditions (*Supplementary file 1w*).

## MRI analyses

### Preprocessing

All MRI data was analysed using SPM12 (Wellcome Trust Centre for Neuroimaging, London, UK) and MATLAB 2021b (The Mathworks Inc 2021b). The brain data was processed using a standardised SPM12-based pipeline (https://github.com/christianbuechel/spm_bids_pipeline, copy archived at *Buechel, 2025*) in BIDS format (*Gorgolewski et al., 2016*) to ensure data accessibility and reproducibility. The T1-weighted anatomical image was acquired on the first of the experimental days. The functional images from both experimental days were pre-processed together. After discarding the first four images of each run (dummy scans), the functional images were slice time corrected and realigned using rigid-body motion correction with six degrees of freedom. Next, a non-linear coregistration was performed using the mean EPI from the realignment and the T1 image. After this, the T1 image was normalised to MNI space (MNI152NLin2009cAsym as provided by CAT12 toolbox) using DARTEL. After this, transformation fields were created by combining the deformation field from the non-linear coregistration with the DARTEL flow fields to map the EPI images to T1 space and template space. A mask for the first-level general linear models (GLMs) was created based on the GM and WM from the T1 image and then warped into the EPI space and finally smoothed with a 3-mm full width at half maximum (FWHM) Gaussian kernel. After this, a second realignment took place but was constrained by a brain mask to minimise the effects of the eye movements. Finally, all EPI images were resliced in their individual space. Noise regressors for the first-level analysis were created using six principal components of white matter responses and six principal components of cerebrospinal fluid (CSF) responses as well as an ROI at the posterior tip of the lateral ventricle (as used in *Horing et al., 2019*). As a final step, careful quality checks took place of all functional images generated after realignment, non-linear coregistration, and normalisation through visual inspection, focusing especially on the successful alignment across both fMRI sessions to allow for subsequent analyses.

## First-level analyses

The single-subject analyses of the brain data were performed in native space using a GLM approach. The design matrices for each subject comprised eight blocks in total, each containing the stimulus onsets for heat and pressure stimuli at all three intensity levels (VAS 30, 50, 70) and a session constant. Additionally, the six motion parameters were augmented by their derivatives (12 parameters) and the squares of parameters and derivatives, resulting in a total of 24 motion parameters (*Friston et al., 1996*). An additional spike correction was performed, where individual volumes with voxels with a deviation of 0.6 mm between each volume with its preceding volume of each run were flagged and individually modelled. In addition, we included RETROICOR (*Glover et al., 2000*) based physiological noise regressors to account for cardiac and respiratory-related motion (*Brooks et al., 2008*). This technique determines cardiac- and respiratory-related noise by allotting cardiac and respiratory phases to individual volumes within a time series. Subsequently, these assigned phases are utilised in a low-order Fourier expansion to generate time-course regressors elucidating signal variations attributed to cardiac or respiratory activities. A refined physiological noise model was applied, computing three cardiac and four respiratory harmonics, along with a multiplicative term incorporating interactions

between cardiac and respiratory noise resulting in 18 regressors (*Harvey et al., 2008*). These resulting noise regressors were included in the first-level analyses. The contrast images from the first-level analysis were then spatially normalised to MNI space using individual deformation fields (i.e. combining non-linear coregistration and DARTEL spatial normalisation) and finally smoothed using a 6-mm FWHM Gaussian smoothing kernel. In addition to a haemodynamic response function model, we also used an FIR model. This model was used to visualise the time course of BOLD responses after the onset of the pain stimulus. In this analysis, the stimulus duration was divided into 12 bins, each covering the duration of one TR spanning a total of 21.6 s after stimulus onset.

## Second-level analyses

Subsequently, the spatially normalised and smoothed contrast images from the first-level analysis were used for random-effects group-level statistics. The successful application of heat pain was established by investigating the parametric contrast heat VAS 70 > 50 > 30, where higher stimulus intensities would recruit a more widespread (sub-)cortical pain network than lower intensities. To explore the effect of drug treatment in the brain, the contrast for the interaction of drug (SAL and NLX) and stimulus intensity (VAS 30, 50, 70) was calculated. Furthermore, we estimated the contrast heat NLX > heat SAL. To investigate the pain modulatory effect of exercise intensity in the SAL condition, the contrasts exercise high (SAL) > exercise low (SAL) and exercise low > high (SAL) were estimated. Furthermore, we extracted the parameter estimates from preregistered opioidergic ROIs (RVM, PAG, frontal midline) and estimated the LMER model corresponding to the behavioural analyses in the SAL condition. To confirm if NLX reduced the hypoalgesic effect, we estimated the contrasts interaction exercise intensity and drug treatment (positive and negative weight) as well as extracting the parameter estimates from the ROIs to conduct LMER models. To investigate the pain modulatory effect of exercise and fitness level in the SAL condition, the contrast exercise high (SAL) > exercise low (SAL) with the mean-centred covariate fitness level (weight-corrected FTP) was estimated. The parameter estimates from each condition were subtracted (LI exercise – HI exercise) and the difference in parameter estimates between both exercise conditions was visualised. To test the three-way interaction of fitness level, sex, and drug in the brain, a two-sample *t*-test of the contrast interaction exercise intensity and drug with the covariate fitness level has been conducted between males and females.

## Correction for multiple comparisons

For whole-brain analyses, we used family-wise-error rate as correction for multiple comparisons, and the according p-values are reported as $p_{corr-WB}$. Furthermore, we used a small volume correction (SVC) mask to correct for multiple comparisons, and the corrected p-values are reported as $p_{corr-SVC}$. This mask was based on the preregistered ROIs rACC, vmPFC (*de la Vega et al., 2016*), PAG (*Bianciardi et al., 2015*; *Bianciardi et al., 2018*; *Singh et al., 2021*; *Singh et al., 2019*; *García-Gomar et al., 2019*; *García-Gomar et al., 2022*), RVM (*Tinnermann et al., 2017*) with an additional 1-mm FWHM Gaussian smoothing kernel (*Figure 7—figure supplement 3* and *Supplementary file 1x*). All thresholds for statistical significance were set to $p < 0.05$. Activation maps at an uncorrected level ($p_{uncorr} < 0.001$) for each contrast are reported in the supplement.

## Acknowledgements

We thank Marilyn Mintah, Marie-Sophie Morgenroth, and Eileen Yawson for their support with data acquisition. We also thank Alexandra Tinnermann for her helpful comments. Further, we thank the radiographers at the Department of Systems Neuroscience and Jürgen Finsterbusch for providing the MR protocol. C.B. and J.N. are supported by ERC-AdG-883892-PainPersist. C.B. is supported by DFG SFB 289 Project A02 (Project-ID 422744262-TRR 289).

---

# Additional information

#### Competing interests
Christian Büchel: Senior editor, eLife. The other authors declare that no competing interests exist.

---

## Funding

| Funder | Grant reference number | Author |
|---|---|---|
| European Research Council | 10.3030/883892 | Janne Ina Nold<br>Tahmine Fadai<br>Christian Büchel |
| Deutsche Forschungsgemeinschaft | SFB 289 Project A02 (Project-ID 422744262-TRR 289) | Christian Büchel |

The funders had no role in study design, data collection, and interpretation, or the decision to submit the work for publication.

## Author contributions

Janne Ina Nold, Conceptualization, Data curation, Software, Formal analysis, Validation, Investigation, Visualization, Methodology, Writing – original draft, Project administration, Writing – review and editing; Tahmine Fadai, Investigation; Christian Büchel, Conceptualization, Resources, Software, Formal analysis, Supervision, Funding acquisition, Methodology, Writing – original draft, Project administration

## Author ORCIDs

Janne Ina Nold ⓘ https://orcid.org/0000-0001-7305-0779
Christian Büchel ⓘ https://orcid.org/0000-0003-1965-906X

## Ethics

All participants gave informed written consent. The study was approved by the Ethics Board of the Hamburg Medical Association (PV7456/2020-10144-BO-ff). We support inclusive, diverse, and equitable conduct of research.

Reviewer #1 (Public review): https://doi.org/10.7554/eLife.102392.3.sa1
Reviewer #2 (Public review): https://doi.org/10.7554/eLife.102392.3.sa2
Author response https://doi.org/10.7554/eLife.102392.3.sa3

# Additional files

## Supplementary files

MDAR checklist

Supplementary file 1. Tables corresponding to the statistical models.

## Data availability

The datasets for the raw behavioural and fMRI data generated in the current study are available on the public repository https://github.com/jannenold/peep_analyses_mri_behavioural. All necessary data to evaluate the results of the study are included in the manuscript and supplementary materials. This study was programmed using MATLAB 2021b and Psychophysics Toolbox (Version 3.0.19). A custom fMRI preprocessing pipeline was used (https://github.com/christianbuechel/spm_bids_pipeline, copy archived at *Buechel, 2025*). The custom behavioural and fMRI analyses pipelines are available on the public repository https://github.com/jannenold/peep_analyses_mri_behavioural (copy archived at *Nold, 2025*). For fMRI data visualisation, we used python-based software Nilearn (https://nilearn.github.io/dev/index.html#; *Thirion et al., 2025*) and an MNI template with 1mm smoothing (MNI152_T1_1mm_brain).

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
