## [Editor Report · eLife Assessment]

In this **valuable** study, Nold et al. examined exercise-induced pain modulation in a pharmacological within-subject fMRI study using the opioid-antagonist naloxone and different levels of aerobic exercise intensity and pain. This investigation provides **solid** evidence to show that the intensity of exercise does not seem to impact the hypoalgesic effect. Moreover, exploratory analysis identified that fitness level and sex may potentially play a role in exercise-induced hypoalgesia, and that further confirmatory studies are required in order to verify these findings.

---

## [Referee Report · Reviewer #1 (Public review)]

Summary:

Participants in this study completed three visits. In the first, participants received experimental thermal stimulations which were calibrated to elicit three specific pain responses (30, 50, 70) on a 0-100 visual analogue scale (VAS). Experimental pressure stimulations were also calibrated at an intensity to the same three pain intensity responses. In the subsequent two visits, participants completed another pre-calibration check (Visit 2 of 3 only). Then, prior to the exercise NALOXONE or a SALINE placebo-control was administered intravenously. Participants then completed 1 of 4 blocks of HIGH (100%) or LOW (55%) intensity cycling which was tailored according to a functional threshold power (FTP) test completed in Visit 1. After each block of cycling lasting 10 minutes, participants entered an MRI scanner and were stimulated with the same thermal and pressure stimulations that corresponded to 30, 50, and 70 pain intensity ratings from the calibration stage. Therefore, this study ultimately sought to investigate whether aerobic exercise does indeed incur a hypoalgesia effect. More specifically, researchers tested the validity of the proposed endogenous pain modulation mechanism. Further investigation into whether the intensity of exercise had an effect on pain and the neurological activation of pain-related brain centres were also explored.Results show that in the experimental visits (Visit 2 and 3), when participants exercised at two distinct intensities as intended. Power output, heart rate, and perceived effort ratings were higher during the HIGH versus LOW intensity cycling. In particular. HIGH intensity exercise was perceived as "hard" / ~15 on the Borg (1974, 1998) scale, whereas LOW intensity exercise was perceived as "very light" / ~9 on the same scale.

The fMRI data from Figure 1 indicates that the anterior insula, dorsal posterior insula and middle cingulate cortex show pronounced activation as stimulation intensity and subsequent pain responses increased, thus linking these brain regions with pain intensity and corroborating what many studies have shown before.

Results also showed that participants rated a higher pain intensity in the NALOXONE condition at all three stimulation intensities compared to the SALINE condition. Therefore, the expected effect of NALOXONE in this study seemed to occur whereby opioid receptors were "blocked" and thus resulted in higher pain ratings compared to a SALINE condition where opioid receptors were "not blocked". When accounting for participant sex, NALOXONE had negligible effects at lower experimental nociceptive stimulations for females compared to males who showed a hyperalgesia effect to NALOXONE at all stimulation intensities (peak effect at 50 VAS). Females did show a hyperalgesia effect at stimulation intensities corresponding to 50 and 70 VAS pain ratings. The fMRI data showed that the periaqueductal gray (PAG) showed increased activation in the NALOXONE versus SALINE condition at higher thermal stimulation intensities. The PAG is well-linked to endogenous pain modulation.

When assessing the effects of NALOXONE and SALINE after exercise, results showed no significant differences in subsequent pain intensity ratings.

When assessing the effect of aerobic exercise intensity on subsequent pain intensity ratings, authors suggested that aerobic exercise in the form of a continuous cycling exercise tailored to an individual's FTP is not effective at eliciting an exercise-induced hypoalgesia response -irrespective of exercise intensity. This is because results showed that pain responses did not differ significantly between HIGH and LOW intensity exercise with (NALOXONE) and without (SALINE) an opioid antagonist. Therefore, authors have also questioned the mechanisms (endogenous opioids) behind this effect.

Strengths:

Altogether, the paper is great piece of work that has provided some truly useful insight into the neurological and perceptual mechanisms associated with pain and exercise-induced modulation of pain. The authors have gone to great lengths to delve into their research question(s) and their methodological approach is relatively sound. The study has incorporated effective pseudo-randomisation and conducted a rigorous set of statistical analysis to account for as many confounds as possible. I will particularly credit the authors on their analysis which explores the impact of sex and female participants' stage of menses on the study outcomes. It would be particularly interesting for future work to pursue some of these lines of research which investigate the differences in the endogenous opioid mechanism between sexes and the added interaction of stage of menses or training status - all of which the authors point out in their discussion.

There are certainly many other areas that this article contributes to the literature due to the depth of methods the research team have used. For example, the authors provide much insight into: the impact of exercise intensity on the exercise-induced hypoalgesia effect; the impact of sex on the endogenous opioid modulation mechanism; and the impact of exercise intensity on the neurological indices associated with endogenous pain modulation and pain processing. All of which, the researchers should be credited for due to the time and effort they have spent completing this study. Indeed, their in-depth analysis of many of these areas provides ample support for the claims they make in relation to these specific questions. As such, I consider their evidence concerning the fMRI data to be very convincing (and interesting).

Weaknesses:

Although the authors have their own view of their results, I, however, do still maintain a slightly different take on what the post-exercise pain ratings seem to show and its implications for judging whether an exercise-induced hypoalgesia effect is present or not and whether this is related to the opioid system.

For example, my basic assumptions relate to data which appears to show that there is an exercise-induced hypoalgesia effect as average pain ratings are ~30% lower than pre-calibrated/resting pain ratings within the SALINE condition at the same temperature of stimulation.Then, it appears there is evidence for the endogenous opioid mechanism as the NALOXONE condition demonstrates a minimal hypoalgesia effect after exercise. I.e., NALOXONE indeed blocked the opioid receptors, and such inhibition prevented the endogenous opioid system from taking effect.

However, through a comprehensive revision of their work, the authors have addressed many areas that myself and my fellow reviewer have questioned and provided a comprehensive set of responses and edits about this. So while I may have some opposing views on the mechanisms at play, I believe that each reader can decide and interpret the data for themselves which has been presented well by the authors.

---

## [Referee Report · Reviewer #2 (Public review)]

Summary:

This interesting study compared two different intensities of aerobic exercise (low-intensity, high-intensity) and their efficacy in inducing a hypoalgesic reaction (i.e. exercise-induced hypoalgesia; EIH). fMRI was used to identify signal changes in the brain, with infusion of naloxone used to identify hypoalgesia mechanisms. No differences were found in post exercise pain perception between the high-intensity and low-intensity conditions, with naloxone infusion causing increased pain perception across both conditions which was mirrored by activation in the medial frontal cortex (identified by fRMI).

Strengths:

The use of fMRI and naloxone provides a strong approach by which to identify possible mechanisms of EIH.The infusion of naloxone to maintain a stable concentration helps to ensure a consistent effect and that the time-course of the protocol won't affect consistency of changes in pain perceptionThe manipulation checks (differences in intensity of exercise, appropriate pain induction) are approached in a systematic way.The interactions for fitness level and sex provide some interesting findings which should be explored further.

Weaknesses:

Given the absence of a baseline/control condition (for exercise), the efficacy of high/low intensity exercise on EIH cannot be assessed. Providing this would have extended and strengthened the findings/conclusions.Whilst the exercise test (functional threshold power) used to set the intensity of the low/high exercise bouts set participants to exercise at different intensities, this method does not ensure that they exercised above/below particular thresholds (i.e. within either heavy or severe domains). This could have created very different relative challenges between participants.

---

## [Author Response]

The following is the authors’ response to the original reviews.

**Public Reviews:**

**Reviewer #1 (Public review):**
Summary:Participants in this study completed three visits. In the first, participants received experimental thermal stimulations which were calibrated to elicit three specific pain responses (30, 50, 70) on a 0-100 visual analogue scale (VAS). Experimental pressure stimulations were also calibrated at an intensity to the same three pain intensity responses. In the subsequent two visits, participants completed another pre-calibration check (Visit 2 of 3 only). Then, prior to the exercise NALOXONE or a SALINE placebo-control was administered intravenously. Participants then completed 1 of 4 blocks of HIGH (100%) or LOW (55%) intensity cycling which was tailored according to a functional threshold power (FTP) test completed in Visit 1. After each block of cycling lasting 10 minutes, participants entered an MRI scanner and were stimulated with the same thermal and pressure stimulations that corresponded to 30, 50, and 70 pain intensity ratings from the calibration stage. Therefore, this study ultimately sought to investigate whether aerobic exercise does indeed incur a hypoalgesia effect. More specifically, researchers tested the validity of the proposed endogenous pain modulation mechanism. Further investigation into whether the intensity of exercise had an effect on pain and the neurological activation of pain-related brain centres were also explored.Results show that in the experimental visits (Visit 2 and 3), when participants exercised at two distinct intensities as intended. Power output, heart rate, and perceived effort ratings were higher during the HIGH versus LOW-intensity cycling. In particular. HIGH intensity exercise was perceived as "hard" / ~15 on the Borg (1974, 1998) scale, whereas LOW intensity exercise was perceived as "very light" / ~9 on the same scale.The fMRI data from Figure 1 indicates that the anterior insula, dorsal posterior insula, and middle cingulate cortex show pronounced activation as stimulation intensity and subsequent pain responses increased, thus linking these brain regions with pain intensity and corroborating what many studies have shown before.Results also showed that participants rated a higher pain intensity in the NALOXONE condition at all three stimulation intensities compared to the SALINE condition. Therefore, the expected effect of NALOXONE in this study seemed to occur whereby opioid receptors were "blocked" and thus resulted in higher pain ratings compared to a SALINE condition where opioid receptors were "not blocked". When accounting for participant sex, NALOXONE had negligible effects at lower experimental nociceptive stimulations for females compared to males who showed a hyperalgesia effect to NALOXONE at all stimulation intensities (peak effect at 50 VAS). Females did show a hyperalgesia effect at stimulation intensities corresponding to 50 and 70 VAS pain ratings. The fMRI data showed that the periaqueductal gray (PAG) showed increased activation in the NALOXONE versus SALINE condition at higher thermal stimulation intensities. The PAG is well-linked to endogenous pain modulation.When assessing the effects of NALOXONE and SALINE after exercise, results showed no significant differences in subsequent pain intensity ratings.When assessing the effect of aerobic exercise intensity on subsequent pain intensity ratings, authors suggested that aerobic exercise in the form of a continuous cycling exercise tailored to an individual's FTP is not effective at eliciting an exercise-induced hypoalgesia response irrespective of exercise intensity. This is because results showed that pain responses did not differ significantly between HIGH and LOW intensity exercise with (NALOXONE) and without (SALINE) an opioid antagonist. Therefore, authors have also questioned the mechanisms (endogenous opioids) behind this effect.Strengths:Altogether, the paper is a great piece of work that has provided some truly useful insight into the neurological and perceptual mechanisms associated with pain and exercise-induced hypoalgesia. The authors have gone to great lengths to delve into their research question(s) and their methodological approach is relatively sound. The study has incorporated effective pseudo-randomisation and conducted a rigorous set of statistical analyses to account for as many confounds as possible. I will particularly credit the authors on their analysis which explores the impact of sex and female participants' stage of menses on the study outcomes. It would be particularly interesting for future work to pursue some of these lines of research which investigate the differences in the endogenous opioid mechanism between sexes and the added interaction of stage of menses or training status.There are certainly many other areas that this article contributes to the literature due to the depth of methods the research team has used. For example, the authors provide much insight into: the impact of exercise intensity on the exercise-induced hypoalgesia effect; the impact of sex on the endogenous opioid modulation mechanism; and the impact of exercise intensity on the neurological indices associated with endogenous pain modulation and pain processing. All of which, the researchers should be credited for due to the time and effort they have spent completing this study. Indeed, their in-depth analysis of many of these areas provides ample support for the claims they make in relation to these specific questions. As such, I consider their evidence concerning the fMRI data to be very convincing (and interesting).Weaknesses:Although the authors have their own view of their results, I do however, have a slightly different take on what the post-exercise pain ratings seem to show and its implications for judging whether an exercise-induced hypoalgesia effect is present or not. From what I have read, I cannot seem to find whether the authors have compared the post-exercise pain ratings against any data that was collected pre-exercise/at rest or as part of the calibration. Instead, I believe the authors have only compared post-exercise pain ratings against one another (i.e., HIGH versus LOW, NALOXONE versus SALINE). In doing so, I think the authors cannot fully assume that there is no exercise-induced hypoalgesia effect as there is no true control comparison (a no-exercise condition).In more detail, Figure 6A appears to show an average of all pain ratings combined per participant (is this correct?). As participants were exposed to stimulations expected to elicit a 30, 50, or 70 VAS rating based on pre-calibration values, therefore the average rating would be expected to be around 50. What Figure 6A shows is that in the SALINE condition, average pain ratings are in fact ~10-15 units lower (~35) and then in the NALOXONE condition, average pain ratings are ~5 units lower (~45) for both exercise intensities. From this, I would surmise the following:It appears there is an exercise-induced hypoalgesia effect as average pain ratings are ~30% lower than pre-calibrated/resting pain ratings within the SALINE condition at the same temperature of stimulation (it would also be interesting to see if this effect occurred for the pressure pain).It appears there is evidence for the endogenous opioid mechanism as the NALOXONE condition demonstrates a minimal hypoalgesia effect after exercise. I.e., NALOXONE indeed blocked the opioid receptors, and such inhibition prevented the endogenous opioid system from taking effect.It appears there is no effect of exercise intensity on the exercise-induced hypoalgesia effect.That is, participants can cycle at a moderate intensity (55% FTP) and incur the same hypoalgesia benefits as cycling at an intensity that demarcates the boundary between heavy and severe intensity exercise (100%FTP). This is a great finding in my mind as anyone wishing to reduce pain can do so without having to engage in exercise that is too effortful/intense and therefore aversive - great news! This likely has many applications within the field of public health.I will very slightly caveat my summaries with the fact that a more ideal comparison here would be a control condition whereby participants did the same experimental visit but without any exercise prior to entering the MRI scanner. I consider the overall strength of the evidence to be solid, with the answer to the primary research question still a little ambiguous.
**Reviewer #2 (Public review):**
Summary:This interesting study compared two different intensities of aerobic exercise (low-intensity, high-intensity) and their efficacy in inducing a hypoalgesic reaction (i.e. exercise-induced hypoalgesia; EIH). fMRI was used to identify signal changes in the brain, with the infusion of naloxone used to identify hypoalgesia mechanisms. No differences were found in postexercise pain perception between the high-intensity and low-intensity conditions, with naloxone infusion causing increased pain perception across both conditions which was mirrored by activation in the medial frontal cortex (identified by fMRI). However, the primary conclusion made in this manuscript (i.e. that aerobic exercise has no overall effect on pain in a mixed population sample) cannot be supported by this study design, because the methodology did not include a baseline (i.e. pain perception following no exercise) to compare high/low-intensity exercise against. Therefore, some of the statements/implications of the findings made in this manuscript need to be very carefully assessed.Strengths:(1) The use of fMRI and naloxone provides a strong approach by which to identify possible mechanisms of EIH.(2) The infusion of naloxone to maintain a stable concentration helps to ensure a consistent effect and that the time course of the protocol won't affect the consistency of changes in pain perception.(3) The manipulation checks (differences in intensity of exercise, appropriate pain induction) are approached in a systematic way.(4) Whilst the exploratory analyses relating to the interactions for fitness level and sex were not reported in the study pre-registation, they do provide some interesting findings which should be explored further.Weaknesses:(1) Given that there is no baseline/control condition, it cannot be concluded that aerobic exercise has no effect on pain modulation because that comparison has not been made (i.e. pain perception at 'baseline' has not been compared with pain perception after high/lowintensity exercise). Some of the primary findings/conclusions throughout the manuscript state that there is 'No overall effect of aerobic exercise on pain modulation', but this cannot be concluded.(2) Across the manuscript, a number of terms are used interchangeably (and applied, it seems, incorrectly) which makes the interpretation of the manuscript difficult (e.g. how the author's use the term 'exercise-induced pain').(3) There is a lack of clarity on the interventions used in the methods, for example, it is not exactly clear the time and order in which the exercise tasks were implemented.(4) The exercise test (functional threshold power) used to set the intensity of the low/high exercise bouts is not an accurate means of demarcating steady state and non-steady state exercise. As a result, at the intensity selected for the high-intensity exercise in this study, it is likely that the challenge presented for the high-intensity exercise would have been very different between participants (e.g. some would have been in the 'heavy' domain, whereas others would be in the 'severe' domain).(5) It is likely that participants did not properly understand how to use the 6-20 Borg scale to rate their perceived effort, and so caution must be taken in how this RPE data is used/interpreted.(6) Although interesting, the secondary analyses (relating to the interaction effects of fitness level and sex) were not included in the study pre-registration, and so the study was not designed to undertake this analysis. These findings should be taken with caution.
**Recommendations for the authors:**

**Reviewer #1 (Recommendations for the authors):**
Participants in this study completed three visits. In the first one, participants received experimental thermal stimulations which were calibrated to elicit three specific pain responses (30, 50, 70) on a visual analogue scale (VAS). Experimental pressure stimulations were also calibrated at an intensity to the same three pain intensity responses. In the subsequent two visits, participants completed another pre-calibration check (Visit 2 of 3 only). Then, prior to the exercise NALOXONE or a SALINE placebo-control was administered intravenously. Participants then completed 1 of 4 blocks of HIGH (100%) or LOW (55%) intensity cycling which was tailored according to a functional threshold power (FTP) test completed in Visit 1. After each block of cycling lasting 10 minutes, participants entered an MRI scanner and were stimulated with the same thermal and pressure stimulations that corresponded to 30, 50, and 70 pain intensity ratings from the calibration stage. Therefore, this study ultimately sought to investigate whether aerobic exercise does indeed incur a hypoalgesia effect. More specifically, researchers tested the validity of the proposed endogenous pain modulation mechanism.Further investigation into whether the intensity of exercise had an effect on pain and the neurological activation of pain-related brain centres was also explored.Results show that in the experimental visits (Visit 2 and 3) when participants exercised at two distinct intensities as intended. Power output, heart rate, and perceived effort ratings were higher during the HIGH versus LOW-intensity cycling. In particular, HIGH intensity exercise was perceived as "hard" / ~15 on the Borg (1974) scale, whereas LOW intensity exercise was perceived as "very light" / ~9 on the Borg (1974) scale.The fMRI data from Figure 1 indicates that the anterior insula, dorsal posterior insula, and middle cingulate cortex show pronounced activation as stimulation intensity and subsequent pain responses increase, thus linking these brain regions with the percept of pain intensity and corroborating what many studies have shown before.Results also showed that participants rated a higher pain intensity in the NALOXONE condition at all three stimulation intensities compared to the SALINE condition. Therefore, the expected effect of NALOXONE in this study seemed to occur whereby opioid receptors were "blocked" and thus resulted in higher pain ratings compared to a SALINE condition where opioid receptors were "not blocked". When accounting for participant sex, NALOXONE had negligible effects at lower experimental nociceptive stimulations for females compared to males who showed a hyperalgesia effect to NALOXONE at all stimulation intensities (peak effect at 50 VAS). Females did show a hyperalgesia effect at stimulation intensities corresponding to 50 and 70 VAS pain ratings. The fMRI data showed that the periaqueductal gray (PAG) showed increased activation in the NALOXONE versus SALINE condition at higher thermal stimulation intensities. The PAG is well-linked to endogenous pain modulation.When assessing the effects of NALOXONE and SALINE after exercise, results showed no significant differences in subsequent pain intensity ratings.When assessing the effect of aerobic exercise intensity on subsequent pain intensity ratings, authors suggested that aerobic exercise in the form of a continuous cycling exercise tailored to an individual's FTP is not effective at eliciting an exercise-induced hypoalgesia response irrespective of exercise intensity. This is because results showed that pain responses did not differ significantly between HIGH and LOW-intensity exercise with (NALOXONE) and without (SALINE) an opioid antagonist. Therefore, authors have also questioned the mechanisms (endogenous opioids) behind this effect.Altogether, the paper is a great piece of work that has provided some truly useful insight into the neurological and perceptual mechanisms associated with pain and exercise-induced hypoalgesia. The authors have gone to great lengths to delve into their research question(s) and their methodological approach is relatively sound. Although the authors have their own view of their results, I do however, have a slightly different take on what the post-exercise pain rating seems to show and its implications for judging whether an exercise-induced hypoalgesia effect is present or not. From what I have read, I cannot seem to find whether the authors have compared the post-exercise pain ratings against any data that was collected preexercise/at rest or as part of the calibration. Instead, I believe the authors have only compared post-exercise pain ratings against one another (i.e., HIGH versus LOW, NALOXONE versus SALINE). In doing so, I think the authors cannot fully question whether there is an exerciseinduced hypoalgesia effect as there is no true control comparison (a no-exercise condition). Nevertheless, there are certainly many other areas that this article contributes to the literature due to the depth of methods the research team has used. For example, the authors provide much insight into: the impact of exercise intensity on the exercise-induced hypoalgesia effect; the impact of sex on the endogenous opioid modulation mechanism; and the impact of exercise intensity on the neurological indices associated with endogenous pain modulation and pain processing. All of which, the researchers should be credited for due to the time and effort they have spent completing this study.I have provided some specific comments for the authors to consider. They are organised to correspond to each section as it is presented, and I have denoted the line I am referring to each time.To conclude, thank you to the authors for their work, and thank you to the editor for the opportunity to contribute to the review of this paper. I hope my comments are seen as useful and I look forward to seeing the authors' responses.

We sincerely appreciate the reviewer's insightful comments, which highlight the strengths of our study. In response to the concerns raised, we have made several key revisions to the original manuscript to address the reviewers’ comments. As for the lack of a resting control condition, we acknowledge that our study was not designed to test the overall effect of exercise versus no exercise. However, our primary objective was to compare different exercise intensities, hypothesising that low-intensity (LI) exercise would induce less pain modulation as compared to high-intensity (HI) exercise. By exploring this, we aimed to enhance understanding of the dose-response relationship between exercise and pain modulation. To better reflect this focus, we have revised the misleading phrasing regarding the ‘overall’ effect of exercise to clearly emphasize our primary aim: comparing HI and LI exercise.

This reviewer suggests an interesting interpretation of the data suggesting that exercise induced hypoalgesia might have occurred for both exercise intensities since the pain ratings provided were lower than the anticipated intensities as determined by the calibration. Given that this difference is lower in the naloxone (NLX) condition could provide evidence of opioidergic mechanisms underlying this effect. Unfortunately, the current study is not designed to comprehensively answer this question since there was no resting control condition. In particular, the lower pain ratings under SAL (Figure 6) could be due to exercise triggering the descending pain modulatory system (DPMS), but equally due to the default activation of the DPMS. Only an additional “no exercise” condition could disentangle this. Furthermore, habituation to noxious stimuli can influence pain ratings, resulting in lower pain ratings during the experiment as compared to the calibration. We have now provided a more detailed overview of the pain ratings at different stimulus intensities after HI and LI exercise in both drug treatment conditions for heat and pressure pain ratings. We elaborated on the specific comments raised in more detail in the following sections.

Specific Comments(1) AbstractLine 25 - "we were unable to"... personal preference but this wording is a little 'weighted' in my view. I personally do not think researchers search to prove hypotheses correct, rather we search to prove hypotheses wrong, and therefore only through repeated attempts of falsification can we surmise that something holds true.

We agree with the reviewer that the chosen wording can be perceived as weighted and have rephrased the sentence.

Line 33 to 35 - the "...but individual factors... might play a role" is a crucial caveat to this sentence for me. Whilst I can understand that the results of the authors' study indicate that prior assumptions about exercise-induced hypoalgesia and its opioidergic mechanisms may be questioned, I think a little more evidence is needed to finally decide whether aerobic exercise has no overall effect on experimental pain responses. (see more in the Results comments below).

We thank the reviewer for their comment. We agree that no claims can be made regarding the effect of aerobic exercise per se on pain modulation compared to no exercise based on the current data. Furthermore, we agree that more research is needed to further advance our understanding of (non-)opioidergic mechanisms in exercise-induced pain modulation. However, based on the data presented in this study we propose that the involvement of endogenous opioids in exercise-induced hypoalgesia could be influenced by sex and fitness levels since we could show differences in opioidergic involvement between males and females of different fitness levels. Future studies should account for the fitness levels and sex of the sample investigated.

(2) IntroductionLine 48 - please predefine anterior cingulate cortex here.

We thank the reviewer for detecting this and have introduced the abbreviation for the anterior cingulate cortex in the referenced line.

Line 49 - please predefine periaqueductal gray here instead of line 52.

We have introduced the abbreviation for periaqueductal grey in the referenced line.

Line 47 to 54 - when discussing the descending pain modulatory systems, authors seem to be relating specifically to the intensity/magnitude of pain experiences. However, the different brain regions that are mentioned may have varying "roles" according to which dimension of pain is of focus.Hofbauer et al. (2001) - https://doi.org/10.1152/jn.2001.86.1.402Rainville et al. (1997) - https://doi.org/10.1126/science.277.5328.968The two above studies provide some nice earlier findings on the brain regions - some of which are mentioned by the authors in this section - associated with the processing of pain quality in addition to the intensity of pain... simply attach here if they are of interest to the authors.

The studies by Hofbauer et al. (2001) and Rainville et al. (1997) provide interesting findings on the effect of hypnotic suggestions on pain affect and the perceived intensity of a painful stimulus. However, these studies did not investigate exercise-induced changes in brain regions of the DPMS. The studies referenced in the relevant section of the manuscript are (one of the few) imaging studies that have indeed investigated brain structures of the DPMS in the context of exercise and pain modulation and, thus, were included in this paragraph to focus on the findings of these studies as well as emphasise the scarcity of imaging studies investigating exercise-induced pain modulation. Given these divergent research topics of the proposed studies, we suggest not including them in this paragraph to maintain a clearer line of argument and focus on exercise-induced pain modulation in brain regions of the DPMS.

L59 to 61 - a minor comment about the phrasing within this sentence and a recommended change is provided below for the flow of the sentence/paragraph."...there are instances where administration of µ-opioid antagonists has decreased exerciseinduced pain modulation (Droste et al. 1988; etc.) whereas in others there has been little effect (Droste et al. 1988; etc.).

We have altered the sentence based on the reviewers' suggestions to improve the flow and coherence of the sentence.

L56 to 72 - Whilst the current version of this paragraph scans well enough, I find that the narrative flits between the mechanisms being discussed and the rationale/shortcomings of current research. I think that the original content of this paragraph can be structured into:A- The endogenous opioid system is a likely candidate to explain how exercise elicits a hypoalgesia response.B- Citation(s) of the imaging studies (Boecker et al., 2008, etc.) and earlier literature which support A (e.g., Janal et al. 1984).C- Further support of this theory as µ-opioid antagonists like naloxone seem to counteract the endogenous opioid effect (Haier et al., 1981).D- Introduction of the caveats of previous research such as the studies that observed that µ-opioids did not impact the endogenous pain modulation system during exercise (e.g., Droste et al., 1991, etc.) and the range of different interventions and exercise modalities which make it difficult to draw clear conclusions of the pain modulation effect.To me, this structure would set out the details you have already put together in a more orderly and systematic way and also will lead nicely into your ensuing paragraph (Line 74 onwards).

We appreciate the reviewers' constructive comments on structuring this paragraph. We agree that the proposed version eases the readability and comprehension of the paragraph and have, thus, adapted the restructured paragraph according to the reviewer’s suggestion.

L75 - Why are single-arm pre-post measures and designs an issue? If you can elaborate a little more this would be very insightful for a reader.

Single-arm pre-post measurement studies involve participants being assigned to a single experimental condition, with pain assessments conducted only once before and once following an intervention. This study design presents some limitations, particularly in the context of examining exercise-induced modulation of pain (Vaegter and Jones, 2020). Such designs are potentially confounded by the effects of habituation to noxious stimuli, as highlighted by Vaegter and Jones (2020). Incorporating randomised controlled trials with multiple measurement blocks not only mitigates these limitations but also provides a clearer understanding of how individual bouts of exercise influence pain perception. We have now added this to the paper.

L80 - The reference for the functional threshold power assessment is provided as a number. Please could the authors change to reflect which study/studies they are referring to here (I presume it is the Borszcz and/or the McGrath studies?).

We apologise for this oversight and have now updated the reference to be displayed correctly. The reviewer is correct in assuming that Borszcz et al. (2018) is the referenced study here.

L88 - Did participants also receive pressure pain stimulations in addition to the thermal stimuli, as the figure suggests?*Note* Since read on to L102-104 and understood why pressure pain was included but not mentioned due to results. However, I would still recommend including pressure pain stimulations in this line, if possible, to be consistent with what Figure 1 shows and later text in the Methods section also shows.

We thank the reviewer for their suggestion to mention pressure pain at the referenced line to increase the clarity and consistency of the experimental paradigm. Pressure and heat pain were applied in alternating fashion during scanning. Whilst the results of pressure pain are not included in this study we agree with the reviewer that it should be mentioned again as part of the methods and have added this.

L94 - I really like Figure 1. Great job.Could the authors please define the inter-trial interval (ITI) in the legend? And please could the authors clarify what unit the 30, 50, and 70 figures in the "18 trials per block" section refer to.

We thank the reviewer for their positive feedback. We have now included a definition of inter-trial-interval (ITI) in the figure legend. Furthermore, we adapted Figure 1 so that the units of the stimulus intensities (30, 50, 70) on the Visual Analog Scale (VAS) are included in the figure allowing for a clearer identification.

(3) ResultsGeneral comment for figures ... is there a specific reason the authors chose for error bars to be represented by an SE value as opposed to an SD value?The reason I ask is that participant responses seem to vary (See Figure 2A and 2E-G as an example). Error bars showing SD values would perhaps do justice to the variability in participant response(s), whereas the SE may be a better representation of the variability in responses due to the assessor's methods of collection. Whilst the SE error bars are narrow (great job on that!), the individual responses are clearly varied which I speculate could be because of the interventions that have been implemented (i.e., exercise intensity).

The use of Standard Error (SE) is more common in the cognitive neuroscience literature.

However, as this reviewer noted, we have also included individual data points alongside the SE, thereby providing a comprehensive view that allows for a thorough interpretation of the data distribution.

L102 to 104 - In fact, it is interesting that exercise did not impact the pressure pain ratings whereas the same cannot be said for thermal pain. In line with some of my comments below about the impact of exercise on pain intensity responses, I would be intrigued to see the results of the pressure pain ratings in more detail.Another note on this... Whilst the results for the pressure pain may be beyond the scope of this paper and will be reported separately, knowing of this data is tantalising for a reader. I would suggest to: (A) either mention the pressure pain and include the analysis of the data; or (B) not mention the pressure pain altogether and save it for the subsequent paper. Either way, I look forward to seeing further discussion on this in future work.

We have now summarised the behavioural results of exercise on pressure pain ratings below in Supplemental Figure S1.

There was no hypoalgesic effect evident in the behavioural pain ratings comparing HI to LI exercise in the saline (SAL) condition (β = 0.57, CI [-1.73, 2.86], SE = 1.17, t(1354) = 0.48, P = 0.63; Supplemental Figure S1A, blue bars) as well as no interaction of drug treatment and exercise intensity on pressure pain ratings (β = -1.43, CI [-4.87, 2.01], SE = 1.75, t(2756.02) = -0.82, P = 0.42; Supplemental Figure S1). Post-hoc paired t-tests (Bonferroni-corrected) confirmed there to be no significant differences between the drug treatment conditions at LI (P = 0.18) or HI (P = 0.85) and no significant difference between the exercise intensities in the SAL (P = 0.65) and NLX (P = 0.48) conditions, confirming no significant differences in drug treatment between the exercise intensities.

Furthermore, there was no significant effect of fitness level on differences in pain ratings (LI – HI exercise) in the SAL condition (β = 3.16, CI [-1.64, 7.97], SE = 2.37, t(38) = 1.34, P = 0.19; Supplemental Figure S1B) and no significant correlation between fitness level and difference pain ratings (r = 0.25, P = 0.13). Finally, there was no significant interaction of drug treatment, exercise intensity, and sex on difference pain ratings (β = -7.97, CI [-18.67, 2.73], SE = 5.51, t(190) = -1.45, P = 0.15; Supplemental Figure S1C-D).

Exercise did not appear to affect pressure pain ratings and we have now added this to the discussion and in the methods section. However, we think that the figure should be part of the supplements.

L112 to 113 - Fantastic work for including this analysis in your study. Great job.

We appreciate the reviewers’ positive feedback on conducting these crucial analyses when investigating sex and gender differences in pain.

L186 to 189 - It is fascinating that there appears to be no effect of NALOXONE on pain ratings within female participants at a VAS rating of 30 for thermal pain as well as a much diminished hyperalgesia effect at a VAS rating of 50 compared to males. Meanwhile, at higher intensity stimulations corresponding to a VAS rating of 70, females in fact demonstrate a more pronounced hyperalgesia effect compared to males. In addition, the hyperalgesia effect of NALOXONE for males seems to "peak" at a VAS rating of 50. The mechanisms behind these findings alone would be incredibly exciting to explore... but maybe in another study.

We agree with the reviewer that the differences in males and females are fascinating results and concur that this may hint at varying degrees of opioidergic involvement at different stimulus intensities. This finding is intriguing and potentially clinically relevant, warranting further investigation in future research, although it lies beyond the scope of the current paper.

L189 - To double check... Figures 4A and 4B refer to the entire cohort (male and female responses combined) whereas C-E are separated by sex?In addition, as there are no annotations to the top of Figures 4C-E were no significant differences observed between saline and naloxone conditions per each stimulus intensity? i.e., similar tests to what are shown in Table S6 but separated for each sex.Without getting too carried away, there may be something here that indicates a difference between sexes concerning the opioid-driven pain modulation response on a neurological level (i.e., brain region activation).

The reviewer is correct in assuming that Figures 4A and 4B refer to the entire cohort whilst Fig. 4C – 4E are split for males and females. The full output of the analyses for Fig. 4A and 4B are reported in Supplemental Tables S5 – S7. Furthermore, the full output of the LMER analyses for Fig. 4E is reported in Supplemental Table S10. We agree with the reviewer that additional annotations in Fig. 4C – Fig. 4E ease interpretation and have, thus, added them to the respective figures, denoting the significance of the interaction term stimulus intensity and drug treatment for females (Fig. 4C) and males (Fig. 4D), respectively. For completeness, we now report the post-hoc paired samples t-tests for females and males in the Supplemental Tables S8 and S9, respectively.

L254 to 258 - "we could not establish an overall hypoalgesia effect of exercise...". Do the results of the exercise intensity x drug treatment provide an answer for this exact hypothesis? After checking the methods section, I cannot seem to find whether the statistical analysis has involved a comparison of the pain ratings after the high (alone), low (alone), or high and low (combined) exercise compared to ratings during control or pre-calibration as part of precalibration (i.e., pain ratings in a rested state without any exercise yet completed).

We concur with the reviewer's assessment that the study design and statistical analyses cannot address the ‘overall’ effect of exercise compared to no exercise. Please refer back to our general response before comment 1, where we have addressed this point.

As it seems that the analysis assesses the differences between high and low-intensity exercise, to me, the results of the exercise intensity x drug treatment analysis do not assess whether there is an exercise-induced hypoalgesia effect or not. Instead, it seems to assess whether the intensity of exercise is a differentiating factor in the expected exercise-induced hypoalgesia effect to subsequent pain intensity ratings to experimental pain stimulation. For the authors to judge whether aerobic exercise does or does not have a hypoalgesia effect, then the exercise conditions (either combined or standalone) would have to be compared to a control condition or a data set that involved pain ratings from a pre-exercise timepoint.

We thank the reviewer for their comment. We would like to point out the we concluded there to be no hypoalgesic effect between the LI and HI exercise based on the LMER model comparing the behavioural pain ratings between the exercise conditions in the SAL condition (β = 1.19, CI [-1.85, 4.22], SE = 1.55, t(1354) = 0.77, P = 0.44; Figure 6A, blue bars and Table S9). The statistical model investigating the interaction of exercise intensity and drug treatment served to show that NLX did not modulate pain differently between the LI and HI exercise conditions.

Given that our experiment involved different exercise levels in a randomized order, a simple pre vs post analysis is not straightforward. Nevertheless, we have set up a model where we take into account the rating time point (pain ratings provided before each exercise block (prepain ratings) and following each exercise block (post-pain ratings)) at each stimulus intensity (VAS 30, 50, 70) and exercise intensity (LI and HI). The model also takes into account the exercise intensity performed in the previous block, the overall block number as well as the varying subject intercepts. The analysis was completed for heat (Author response image 1A) and pressure (Author response image 1B) pain ratings in the SAL condition to establish whether there was a significant effect of exercise intensity on the changes from pre to post-pain ratings. The model for heat pain yielded a significant main effect for stimulus intensity (β = 1.43, CI [1.34, 1.52], SE = 0.05, t(2054.95) = 31.61, P < 0.001) but no significant interaction of exercise intensity, rating time point, and stimulus intensity (P = 0.14). The model for pressure pain in the SAL condition yielded a significant main effect of stimulus intensity (β = 1.00, CI [0.92, 1.08], SE = 0.04, t(2054.99) = 24.68, P < 0.001) and block number (β = 1.14, CI [0.35, 1.94], SE = 0.41, t(2055.98) = 2.80, P = 0.005) but not interaction of exercise intensity, rating time point, and stimulus intensity (P = 0.38).

**Author response image 1. sa3fig1:** Heat (A) and Pressure (B) pain ratings in the saline (SAL) condition for pre (purple) and post (turquoise) exercise pain ratings at LI and HI exercise and all stimulus intensities (VAS 30, 50, 70). The bars depict the mean pain rating pre and post-exercise and the dots depict the subject-specific mean ratings. The error bars depict the SEM.

Another point of consideration is that Figure 6A appears to show an average of all pain ratings combined per participant (is this correct?). As participants were exposed to stimulations expected to elicit a 30, 50, or 70 VAS rating based on pre-calibration values, therefore the average rating would be expected to be around 50. What Figure 6A shows is that in the SALINE condition, average pain ratings are in fact ~10-15 units lower (~35) and then in the NALOXONE condition, average pain ratings are ~5 units lower (~45) for both exercise intensities. From this, I would surmise the following:It appears there is an exercise-induced hypoalgesia effect as average pain ratings are ~30% lower than pre-calibrated/resting pain ratings within the SALINE condition at the same temperature of stimulation (it would also be interesting to see if this effect occurred for the pressure pain).It appears there is evidence for the endogenous opioid mechanism as the NALOXONE condition demonstrates a minimal hypoalgesia effect after exercise. I.e., NALOXONE indeed blocked the opioid receptors, and such inhibition prevented the endogenous opioid system from taking effect.It appears there is no effect of exercise intensity on the exercise-induced hypoalgesia effect. That is, participants can cycle at a moderate intensity (55% FTP) and incur the same hypoalgesia benefits as cycling at an intensity that demarcates the boundary between heavy and severe intensity exercise (100%FTP). This is a winner in my mind. Anyone wishing to reduce pain can do so without having to engage in exercise that is too effortful and therefore aversive - great news!I will very slightly caveat my summaries with the fact that a more ideal comparison here would be a control condition whereby participants did the same experimental visit but without any exercise prior to entering the MRI scanner.As a result of this interpretation of your findings, I do not think that aerobic exercise as a means to cause subsequent hypoalgesia to experimental thermal nociception can be fully discounted. On the contrary, I think your results showed in Figure 6A are evidence for it.

The reviewer is correct in assuming that Figure 6A shows the averaged pain ratings across all stimulus intensities (VAS 30, 50, and 70) for each subject. To provide more details, we have split Figure 6A by stimulus intensity, now depicting the pain ratings for LI and HI exercise and treatment condition (SAL and NLX) at VAS 30, 50, and 70 (Supplemental Fig. S8). The LMER was extended to include the stimulus intensity and yielded a significant main effect of stimulus intensity (β = 1.39, CI [1.31, 1.47], SE = 0.04, t(2753.12) = -34.082, P < 0.001) and a significant interaction of stimulus intensity and drug treatment (β = 0.12, CI [0.01, 0.24], SE = 0.06, t(2751) = 2.13, P = 0.03) but no significant interaction of exercise intensity, drug treatment, and stimulus intensity (β = -0.05, CI [-0.20, 0.11], SE = 0.08, t(2751) = -0.56, P = 0.58).

The reviewer further suggests that the average pain ratings in the SAL condition are lower than the anticipated stimulus intensity, thus, indicating exercise-induced hypoalgesia. While this interpretation is one possibility, there is an alternative explanation: the lower pain ratings may stem from habituation to heat pain (Greffrath et al., 2007; Jepma et al., 2014; May et al., 2012). To support this perspective, we have visualised data from other studies in our lab that have been conducted with the same thermode head and device (TSA-2), using the same calibration procedure and aiming for the same stimulus intensities (VAS 30, 50, and 70). In both studies (Author response image 2A: Study 1: Behavioural sample; Author response image 2B: Study 2: fMRI sample; Author response image 2C: Original Exercise Study), participants did not engage in an exercise task and the pain ratings at VAS 30 and VAS 50 were lower than the anticipated intensities (VAS 30: 11.1/13.4; VAS 50: 35.0/35.9). Furthermore, in a previous study by (Wittkamp et al., 2024), the authors showed that, despite calibrating the heat stimuli at VAS 60, participants rated the pain stimuli with M = 48.58 (SD = 13.79).

This discrepancy observed between calibrated intensities and ratings provided could be attributable to habituation effects, especially at low-intensity stimuli. Moreover, we would like to point the reviewer to the highest stimulus intensity at VAS 70 (Author response image 2C), where no habituation in all three data sets (including the current study) has taken place. This consistency suggests that exercise-induced hypoalgesia may not be present in our findings or potentially confounded by habituation effects.

**Author response image 2. sa3fig2:** Heat pain ratings at different intensities (30, 50, and 70 VAS) in different study samples. Bars depict the mean ratings in the saline (SAL) condition. Individual data points depict subject-specific mean pain ratings. Error bars depict the SEM.

The reviewer further suggests that there is evidence for endogenous opioidergic modulation since the pain ratings in the NLX condition are lower than the anticipated intensities. We fully agree but, again, would argue that the DPMS can exert its effects on painful stimuli in a default manner, i.e. irrespective of any exercise effect.

We concur with the reviewer’s interpretation that there is no effect of exercise intensity on exercise-induced hypoalgesia since the ratings between both exercise intensities are not significantly different.

Finally, we agree that our data does not allow for the interpretation of an ‘overall’ effect of exercise-induced hypoalgesia and would like to point out that we did not aim to claim this. Rather, the data suggests there to be no effect of LI vs. HI aerobic exercise on pain modulation. We acknowledge, however, that the phrasing involving ‘overall’ can be misleading and have revised this to focus on the comparison between LI and HI exercise, thereby enhancing precision and clarity.

*Note* This is also where it would be really interesting to see the pain pressure data if it were to be included. Mainly to see whether it coheres with what the thermal stimulation stuff shows.

We have provided the ratings for the pressure pain ratings in the SAL condition below (Author response image 3).

**Author response image 3. sa3fig3:** Pressure pain ratings in the SAL condition at stimulus intensity (VAS 30, 50, and 70). Bars depict the mean ratings in the saline (SAL) condition. Individual data points depict subject-specific mean pain ratings. Error bars depict the SEM.

L259 - As mentioned in the comment above. Could the authors distinguish what is being shown in Figure 6A? Are the data presented as the pooled mean for all stimulation intensities? If not, what data is displayed per bar/column?

We thank the reviewer for their comment. The reviewer is correct in assuming that the bars in Figure 6A depict the pooled means across all stimulus intensities (VAS 30, 50, 70) for each drug treatment condition and exercise intensity. To allow for a more detailed comprehension of the data, we have split Figure 6A by stimulus intensity, now depicting the pain ratings for LI and HI exercise and treatment condition (SAL and NLX) at VAS 30, 50, and 70 (Supplemental Figure S8). The LMER was extended to include the stimulus intensity and yielded a significant main effect of stimulus intensity (β = 1.39, CI [1.31, 1.47], SE = 0.04, t(2753.12) = -34.082, P < 0.001) and a significant interaction of stimulus intensity and drug treatment (β = 0.12, CI [0.01, 0.24], SE = 0.06, t(2751) = 2.13, P = 0.03) but no significant interaction of exercise intensity, drug treatment, and stimulus intensity (β = -0.05, CI [-0.20, 0.11], SE = 0.08, t(2751) = -0.56, P = 0.58).

L278 - Can the authors please provide a reference that explains how W.kg-1 at FTP is a measure of fitness level?

We thank the reviewer for their comment. The obtained FTP value was corrected for the weight of each participant (Watt/kg), yielding a weight-corrected fitness measure that allows for better comparison between subjects. We denoted this in the figures as W*kg-1 which serves to be the equivalent term.

L296 - Take the line away from Figure 7A... Does the individual data show a positive relation between pain rating changes and W.kg-1? Besides the three data points (1 on the far right of the figure and the two on the far left), I find it hard to see any real trend.

We acknowledge the reviewers’ concern regarding the regression line and the visual clarity of the individual data points. However, it is important to note that the significant main effect of fitness level on differences in pain ratings in the SAL condition (β = 6.45, CI [1.25, 11.65], SE = 2.56, t(38) = 2.52, P = 0.02) supports the assertion that higher fitness levels are associated with greater hypoalgesia following HI exercise compared to LI exercise. While the trend may not be visible for all data points, the statistical analysis provides a robust basis for the observed relationship (r = 0.33, P = 0.038).

We have conducted an additional LMER model where we have excluded the subjects with the highest and lowest FTP values (sub-28 with 3.19 W/kg and sub-06 with 0.76 W/kg, respectively.) The LMER still yields a significant main effect of fitness level (β = 6.82, CI [1.25, 11.65], SE = 3.18, t(34) = 2.14, P = 0.039; Author response image 4) and a positive correlation between the difference ratings and fitness level approaching significance (r = 0.32, P = 0.057).

**Author response image 4. sa3fig4:** Fitness level on difference pain ratings (LI-HI exercise) without subjects with highest and lowest FTP (N = 37). (A) Subject-specific differences in heat pain ratings (dots) between LI and HI exercise conditions (LI – HI exercise pain ratings) and corresponding regression line pooled across all stimulus intensities in the SAL condition. Fitness level (FTP) showed a significant positive relation to heat pain ratings with a significant main effect of FTP (P = 0.039) on difference ratings.

(4) DiscussionL356 to 358 - Exactly. What you write here, I agree with. Your testing allowed you to judge whether there is an effect of aerobic exercise intensity on pain modulation. However, I think this has been a little conflated with the idea that there is "no overall effect of aerobic exercise on pain modulation" in other areas of the article (L358-361, Results, and Abstract). As per my previous comment, I am not sure this (no overall effect) is true.

We agree with the reviewer and have adapted the manuscript so that the misleading phrase including ‘overall’ is removed.

L358 to 365 - One addition to this debate about whether this is a hypoalgesia effect of aerobic exercise. In 358 - 361 (particularly the end of 361) there is a strong conclusion that there is no direct involvement of the endogenous opioid system. Then glance onto L364 to 365 and there is then an almost conflicting summary that a hypoalgesia effect driven by opioidergic regions of the brain (and ergo endogenous opioids) is in effect. If there were no direct endogenous opioid involvement, then differences between NALOXONE (blockade of the opioid mechanism) and SALINE conditions would not exist.

We thank the reviewer for their comment. The structure of this paragraph aimed to guide the reader towards a more nuanced understanding of the possible mechanisms and caveats in exercise-induced pain modulation. Whilst our data suggest an effect of NLX on pain ratings where we showed significantly higher pain ratings in the NLX condition compared to the SAL condition we could not identify an interaction between treatment and exercise intensity. This suggests that there is no significant difference in opioidergic involvement between HI and LI exercise. Our exploratory analyses, however, show an effect of endogenous opioids involved as an underlying mechanism dependant on sex and fitness level.

My perspective is that an exercise-induced hypoalgesia effect has occurred (based on the data in Figure 6A) but that this effect is certainly caveated by the sex and fitness levels that this study has observed (and kudos for it).

As mentioned above, based on the current data we cannot untangle whether the reduced pain ratings in the SAL condition are due to habituation to noxious stimuli or an actual hypoalgesic effect of exercise (or potentially a mix of both). However, we fully agree with the reviewer that exercise-induced pain modulation is influenced by fitness level and sex.

L390 - "endogenous pain modulation through μ-opioid receptors increases with increasing pain intensity". Aside from the general discussion about whether aerobic exercise causes a post-exercise hypoalgesia effect. This finding is also interesting for the pain incurred during exercise in the form of naturally occurring muscle pain and may also be clinically relevant as it could be that the endogenous pain modulation "system" could be primed through repeated exercise as your results show that the fitness level (i.e., a close correlate of how much someone has engaged in exercise and therefore 'activated' the endogenous pain modulation system) is associated with a more pronounced post-exercise hypoalgesia effect.

This is an interesting aspect. With regards to the pain induced by exercise itself (i.e. muscle pain) we did not gather any data on this type of pain and interpreting this would be mere speculation. However, it is an interesting hypothesis to investigate in future studies whether the pain induced by exercise is potentially influenced by the endogenous opioid system. We agree with the reviewers’ interpretation that repeated exercise might prime the endogenous opioid system, especially in fitter individuals who engage more frequently in exercise and, thus, ‘train’ the endogenous opioid system. We have included this line of interpretation in the original manuscript, where we suggest that the mFC, a brain region with high µ-opioid receptor density, might be ‘trained’ by repeated exercise and, therefore, shows increase activation in fitter individuals after short bouts of exercise.

L404 to 405 - "a resting baseline does not control for unspecific factors such as attentional load or distraction (Brooks et al., 2017; Sprenger et al., 2012) through exercise." I am not sure I agree. A control condition allows one to truly deduce whether exercise causes a hypoalgesia effect or not. The attentional load may be a factor, but I would argue this is distinct from endogenous pain modulation - unless there is a study that shows cognitive load alone can elicit endogenous opioids like exercise. About distraction, this would be the case if the pain measures were taken during the exercise. However, as the pain measures taken in the MRI were post-exercise and there was no added distraction related to the exercise present anymore, then I do not think any added effect of distraction due to the exercise and its effect on postexercise pain measure is relevant any longer.

We agree with the reviewer that a resting baseline condition in the context of exercise induced pain modulation would allow for the investigation of a potential hypoalgesic effect of exercise compared to no exercise. It is important to note that both studies (Brooks et al., 2017; Sprenger et al., 2012) have indeed shown that the effect of cognitive pain modulation is mediated by endogenous opioids.

L406 - I do not think a low-intensity exercise is a true "control" condition. It certainly does allow the study to compare the dose-response relationship but as the individual is exercising (even at a moderate physiological intensity) then comparison of HIGH vs LOW does not tell us whether exercise does or does not cause hypoalgesia. In contrast, the results from Figure 6A seem to show that even LOW intensity exercise has a hypoalgesia effect and this is a good thing for those who cannot exercise at high intensities (e.g., chronic populations).

Please refer back to our general response before comment 1, where we have addressed this point.

L410 - A small digression in relation to the exercise intensities:The intensity domains (moderate - heavy - severe) are not truly controlled within this study (mainly for the LOW condition), and therefore some participants could have exercised within different exercise intensity domains than others. To explain, the exercise intensity domains are distinguishable by the physiological responses associated with the boundaries of each of these domains. The FTP is believed to be a demarcation point between heavy and severe intensity domains (though kinesiologists debate the validity of this). Other concepts similar to FTP are Critical Power or the Respiratory Compensation Point. Ultimately, the boundary between heavy and severe intensity domains is characterised by the highest possible intensity by which a steady-state in oxygen kinetics (V̇ O2) occurs (Burnley & Jones, 2018). If this is expressed as a power output (Watts) and then a percentage of this power output is used to prescribe exercise intensity, then the physiological response is not always as expected. The reason is that for some people the gaseous exchange threshold (the demarcation point between the moderate and heavy intensity domains) is not always the same percentage between resting and FTP/Critical Power/Respiratory Compensation Point for each person. As a result, some individuals who are prescribed an intensity of 55% FTP/Critical Power/Respiratory Compensation Point may subsequently exercise within the moderate intensity domain (most people did based on the heart rate and RPE responses) whilst some others might actually exercise more within the heavy intensity domain. A quick check of Figures 3B-C could indicate that this might have been the case for two or three participants, but that is inference and speculation as we cannot truly know unless gas parameters were taken (which is perfectly understandable that they have not been taken because this study has done so much else). However, the importance of this for this study is that if some participants did indeed exercise at a slightly higher physiological intensity, this undermines the LOW condition as a "control" as the physiological stimulus between conditions (Brownstein et al., 2023). It means that the proposed differences in endogenous opioids (Vaegter et al., 2015; 2019) between exercise intensities may not have been present and therefore summarising a lack of an exercise induced hypoalgesia effect is slightly confounded. This is one factor contributing to my scepticism about the conclusion that there is a lack of an exercise-induced hypoalgesia response.

We thank the reviewer for their comment as it touches upon the challenges of estimating exercise intensities precisely. It is, indeed, crucial to consider the boundaries between moderate, heavy, and severe intensity domains, as delineated by physiological markers such as the Functional Threshold Power (FTP), Critical Power, and the Respiratory Compensation Point (VO2max) (Burnley & Jones, 2018). Previous research has shown that the FTP and FTP20 tests are reliable and convenient methods to estimate approximate measures of VO2max (Denham et al., 2020) and that the FTP test is a useful test for performance prediction in moderately trained cyclists (Sørensen et al., 2019).

We acknowledge that without direct measurements of VO2max, it is challenging to determine the precise intensity domain in which each participant was operating. While the RPE and HR might suggest that some participants performed in the moderate intensity domain in the LI exercise condition, we could still ascertain there to be a significant difference in the relative power (%FTP), heart rate (HR), and rating of perceived exertion (RPE) between the LI and HI exercise conditions. In the overall sample, the consistency in relative power, heart rate, and RPE responses among participants suggests that the exercise doses were effectively communicated and adhered to; therefore, the validity of the LI exercise condition remains robust.

While we did not include metabolic assessments in our protocol, our study focused on providing a comprehensive analysis of the exercise-induced hypoalgesia phenomenon across two distinct exercise intensities. Additionally, the rationale for selecting specific exercise intensities was grounded in the existing literature, which indicates significant differences in the hypoalgesic response between exercise intensity levels (Jones et al., 2019; Vaegter et al., 2014).

According to the reviewer, the potential lack of difference between the exercise conditions might contribute to the fact that there was no difference in endogenous opioid release and, thus, no difference in pain ratings between the exercise conditions. However, our data still suggests that there is an influence of endogenous opioids in the HI exercise condition in males with higher fitness levels. Together with recent findings on the association of µ-opioid receptor activation and fitness levels in men (Saanijoki et al., 2022), as well as the difference in µ-opioid receptor availability between high and moderate aerobic exercise (Saanijoki et al., 2018), we would hypothesise that the release of endogenous opioids after short HI bouts of exercise depend on fitness levels (and potentially sex).

Finally, we propose that discussing exercise intensity domains within the context of our study enriches the understanding of exercise-induced hypoalgesia without undermining the integrity of our findings. We have, therefore, included this in the discussion of the manuscript.

L417 - For some reason I am doubting this value (r = 0.61). Could this be checked? I think it is higher in their study. r = 0.88?Also, as someone with a kinesiology background, I would argue this is a given anyway. The maximum power one can cycle for 20 minutes is related to the maximum power one can cycle for 60 minutes, this is expected. (That is no slight on the authors of this study, more a remark that readers could look and figure that for themselves if they needed to know).

We thank the reviewer for their comment. We have carefully re-checked the correlation coefficient between the FTP20 and FTP60 tests in the study by Borsczc et al. (2018) and have corrected the correlation coefficient to r = 0.88. We thank the reviewer for detecting this. Whilst we agree that it seems somehow intuitive that the FTP20 and FTP60 should correlate highly, we wanted to provide the reader with a better understanding of where the FTP20 tests originated from and how it is suitable to assess aerobic fitness levels without having to maintain a steady power output for 60 minutes.

L428 - Kudos to the authors for taking a standardised approach to this. Hopefully, my comment earlier might provide some extra food for thought about exercise intensity. I think there are several other ways future research could prescribe exercise without the need for expensive and cumbersome bits of equipment to know how hard people are exercising.

We strongly agree with the reviewer and hope that our study can inspire future research to implement more convenient and inexpensive ways to establish aerobic (and anaerobic) fitness levels.

L456 to 458 - Would it be possible to revisit this and check whether the pooled mean of all stimulation intensities for pain intensity ratings after pressure pain is lower than 50? If so, I think it can also be assumed that there is a slight hypoalgesia effect occurring for pressure pain too.

We have revisited the pressure pain ratings pooled across all stimulus intensities (VAS 30,50, and 70). Indeed, the ratings are below 50 VAS (Supplemental Figure S1A) in the SAL and NLX conditions. As mentioned before lower pain ratings after LI exercise cannot be taken as evidence for exercise-induced analgesia.

L495 to L499 - I find this fascinating. Great finding.

We thank the reviewer for their positive feedback.

(5) MethodsL650 - "Watts"

We have changed the sentence accordingly.

L651 - beats per minute can also be represented as b.min-1 and cadence as revolutions.min-1.

To allow for easier interpretation of the results in a broader readership we would like to propose to maintain the original abbreviations.

L678 - Just to check what the authors mean by "on the second experimental day", they are actually referring to Visit 2 of 3 (first experimental visit of 2) as it is shown in Figure 1?

We apologise for the lack of clarity. Indeed, the second experimental day refers to the third visit in the study. We have added this to the sentence to increase clarity.

L708 - would change the end of the sentence to "and remained blinded throughout the study"

We have changed the sentence accordingly.

L742 - comma after "in one participant".

We have added the missing comma.

L746 - slight mistype... RPE in brackets instead of PRE

We have changed the abbreviation to RPE.

L747 - In case the authors are interested in affective measures in future studies... Hardy and Rejeski (1989) have a 9-point Likert scale rating affective valence which might be useful to check out.

Thank you. The scale by Hary and Rejeski (1989) is a very relevant measure of affective valence during exercise, and we will consider this in future studies.

L755 - Four squares for the thermode to be applied were drawn on the arm but through the methods I can only seem to see that the thermode was applied to the second square during calibration. During the MRI scan, did someone move the thermode to different squares for different stimulations?

We appreciate the reviewers' question. Indeed, the heat calibration and recalibration on the first and second day, respectively, have always been completed on the same skin patch (patch 2) to allow for comparability of calibration across days. During the experimental sessions, the thermode head was repositioned in a randomised order across participants (i.e., skin patch 14-3-2) before each block. This was done manually before the MRI block commenced. The order of thermode head position was kept constant within participants across experimental days (day 2 and day 3).

L764 - ITI predefined?

We thank the reviewer for their comment and would like to point to line 130 in the revised manuscript where the abbreviation for inter-trial-interval (ITI) was first introduced.

(6) Other Sections + Supplementary MaterialsL891 - I apologise in advance for this comment as it is the most trivial comment you will ever receive, but there is an extra "." On this line after J.N. initials for methodology.

We have changed the punctuation accordingly.

Table S1 - Strictly speaking, some of the intensity denominations in this table are not exactly an "intensity".Iannetta et al. (2020) - https://doi.org/10.1249/mss.0000000000002147 provides a commentary on intensity domains as well as Burnley and Jones (2018) - https://doi.org/10.1080/17461391.2016.1249524Likewise in this table - the term "without fatigue" in the description column is not strictly true as participants will naturally fatigue but authors are referring more to a "steady state".

We have changed the name of the column to ‘Description’ to describe the test phase as proposed by Allen and Coggen (2012) and previously implemented by McGrath et al. (2019) and not the ‘intensity domains’ (as specified by Iannetta et al. (2020)). Further, we have refined the wording in Table S1 and replaced the term ‘without fatigue’ with ‘steady state’.

Once again, thank you to the authors for their great work on this project and to the editor for the chance to review this paper.

We would like to thank this reviewer for their very insightful and important comments and for pointing out the strengths of the manuscript. We believe the suggestions will help to improve the quality of the manuscript.

**Reviewer #2 (Recommendations for the authors):**
Summary:This interesting study compared two different intensities of aerobic exercise (low-intensity, high-intensity) and their efficacy in inducing a hypoalgesic reaction (i.e. exercise-induced hypoalgesia; EIH). fMRI was used to identify signal changes in the brain, with the infusion of naloxone used to identify hypoalgesia mechanisms. No differences were found in postexercise pain perception between the high-intensity and low-intensity conditions, with naloxone infusion causing increased pain perception across both conditions which was mirrored by activation in the medial frontal cortex (identified by fMRI). However, the primary conclusion made in this manuscript (i.e. that aerobic exercise has no overall effect on pain in a mixed population sample) cannot be supported by this study design, because the methodology did not include a baseline (i.e. pain perception following no exercise) to compare high/low-intensity exercise against. Therefore, some of the statements/implications of the findings made in this manuscript need to be very carefully assessed.Strengths:(1) The use of fMRI and naloxone provides a strong approach by which to identify possible mechanisms of EIH.(2) The infusion of naloxone to maintain a stable concentration helps to ensure a consistent effect and that the time course of the protocol won't affect the consistency of changes in pain perception.(3) The manipulation checks (differences in intensity of exercise, appropriate pain induction) are approached in a systematic way.(4) Whilst the exploratory analyses relating to the interactions for fitness level and sex were not reported in the study pre-registation, they do provide some interesting findings which should be explored further.Weaknesses:(1) Given that there is no baseline/control condition, it cannot be concluded that aerobic exercise has no effect on pain modulation because that comparison has not been made (i.e. pain perception at 'baseline' has not been compared with pain perception after high/low intensity exercise). Some of the primary findings/conclusions throughout the manuscript state that there is 'No overall effect of aerobic exercise on pain modulation', but this cannot be concluded.(2) Across the manuscript, a number of terms are used interchangeably (and applied, it seems, incorrectly) which makes the interpretation of the manuscript difficult (e.g. how the author's use the term 'exercise-induced pain').(3) There is a lack of clarity on the interventions used in the methods, for example, it is not exactly clear the time and order in which the exercise tasks were implemented.(4) The exercise test (functional threshold power) used to set the intensity of the low/high exercise bouts is not an accurate means of demarcating steady state and non-steady state exercise. As a result, at the intensity selected for the high-intensity exercise in this study, it is likely that the challenge presented for the high-intensity exercise would have been very different between participants (e.g. some would have been in the 'heavy' domain, whereas others would be in the 'severe' domain).(5) It is likely that participants did not properly understand how to use the 6-20 Borg scale to rate their perceived effort, and so caution must be taken in how this RPE data is used/interpreted.(6) Although interesting, the secondary analyses (relating to the interaction effects of fitness level and sex) were not included in the study pre-registration, and so the study was not designed to undertake this analysis. These findings should be taken with caution.

We thank the reviewer for their insightful comments that contribute to improving the quality of the manuscript. In response to the identified weaknesses, we have made key revisions to enhance clarity and rigor. Regarding the lack of a resting control condition, we acknowledge that our study does not assess the overall effect of exercise versus no exercise. Our primary objective was to compare high- (HI) and low-intensity (LI) exercise on pain modulation, hypothesizing that lower intensities would have minimal effects. We revised the manuscript to eliminate misleading phrases about an "overall" effect, clearly emphasizing our aim to investigate the comparative effects of different exercise intensities. To address terminology inconsistencies, we have adopted "exercise-induced pain modulation," reflecting existing literature that recognizes both hypoalgesia and hyperalgesia associated with exercise (Vaegter and Jones, 2020). We clarified this terminology in the introduction and specified the pain modalities used in our study. We also improved methodological transparency by better describing the timing and order of exercise and drug treatment interventions. Concerning exercise intensity estimation, we acknowledge the complexities in classifying moderate, heavy, and severe domains. We added the study by Wong et al. (2023) to discuss the potential limitations of the FTP estimation protocol. Although direct measures of VO2max or blood lactate are absent in our study, our findings, including perceived exertion (RPE) scores and relative power data, support that participants were primarily in the heavy-intensity domain during HI exercise. To clarify RPE ratings, we adjusted the presentation to align with the Borg scale's intended anchor points, ensuring greater accuracy in reported exertion levels. Statistical analyses confirm significant differences in RPE between exercise intensities. These revisions aim to clarify our intent and methodologies, ultimately strengthening the contribution of our research to understanding exercise-induced pain modulation.

(1) Lines 27-33 - please present some data and accompanying statistical output in the results section of the abstract.

We thank the reviewer for their comment. In the results section of the abstract, we report whether the findings are (not) significant using the general threshold of P < 0.05. However, we prefer not to include more detailed data and statistical outputs here, as these are thoroughly presented in the results section and do not contribute to the abstract’s primary purpose of providing a concise summary.

(2) Line 29 - please indicate how fitness level was quantified.

The functional threshold power (FTP) adjusted for weight served as an indication of cardiovascular fitness level. We have now included this in the abstract.

(3) Line 35 - please include a sentence detailing the implications of your findings.

We have now included a sentence on the implications of our findings in the abstract.

(4) Introduction general - I appreciate that it was an exploratory analysis, however, the introduction does not particularly lay the groundwork for this (e.g., the influence of fitness level, sex, etc) - please include some background within the introduction to establish the role level of fitness/exercise/training/physical activity on pain modulation.

A paragraph detailing the role of fitness level and sex in the context of exercise-induced pain modulation and endogenous opioid release was part of the introduction of our manuscript but has been removed as per the reviewing editor’s request (as the inclusion of sex and fitness level was not part of the preregistration). We have now re-included a shortened version of this paragraph to provide some background on these potentially crucial factors in exercise-induced pain modulation.

(5) Lines 40-41 - reference needed.

We thank the reviewer for detecting this and have now included references concerning the release of endogenous opioids and the term exercise-induced hypoalgesia.

(6) Lines 48-49 - please provide the full terms for ACC and PAG (PAG has been provided on line 52, but should be presented earlier).

We thank the reviewer for detecting this. We now introduce the abbreviations for the periaqueductal grey (PAG) and anterior cingulate cortex (ACC) in the correct lines.

(7) Line 49 - the term exercise-induced pain is often used interchangeably (incorrectly) with many different types of pain experienced during/after exercise (e.g. muscle burn/ache, DOMS, injury etc.). Please see O'Malley et al 2024 (doi: 10.1113/EP091687).

We thank the reviewer for their comment. Despite the distinction between different types of pain induced by exercise being important, this is less relevant for the current study. We would like to point out that the full term used is exercise-induced pain modulation, referring to the modulation of (experimental) pain through exercise. We have deliberately chosen this term as it summarises exercise-induced hypoalgesia as well as hyperalgesia. Therefore, we did not refer to pain induced by exercise and would disagree that this term has been used interchangeably with different types of pain in the current manuscript.

(8) Line 57 - neither of these studies looked at exercise-induced pain, rather they examined experimentally induced pain (e.g. cold pressor test) or chronic pain and how exercise might exacerbate it. This leads back to the previous comment - it is important to define what is meant by exercise-induced pain (EIP) from the offset, and then remain consistent in the reference to this.

We agree with the reviewer and have cited the studies accordingly. We would like to point out that the current study does not investigate exercise-induced pain but the modulation of experimental pain through exercise and have used the term exercise-induced pain modulation consistently in the manuscript to describe this.

(9) Line 61 - Droste et al and Olausson et al are missing from the reference list.

We apologise for this oversight and have now updated the reference list to include the studies by Droste et al. (1991) and Olaussen et al. (1986).

(10) Line 61 - Do you mean exercise-induced hypoalgesia, or modulation of exercise-induced pain - it is not clear? EIH is introduced in Line 40 and in consistent with what the Koltyn study explored. Conversely, Koltyn induced pain using heat and pressure, rather than exercise.

In this manuscript, we have opted for the term ‘exercise-induced pain modulation’ since previous research has shown that exercise can elicit hypoalgesia as well as hyperalgesia (for review see Vaegter and Jones (2020)). Thus, the term refers to the modulation of pain through exercise. We have now included a sentence detailing the use of the term ‘exercise-induced pain modulation’ in the first passage of the introduction. Corresponding to Koltyn et al. (2014), we have used heat and pressure stimuli to induce pain and investigate the modulating effect of different exercise intensities on these pain modalities.

(11) Line 62 and 64 - Both the Janal study and Haier study are missing from the reference list.

We apologise for this oversight and have now updated the reference list to include the studies by Janal et al. (1984) and Haier et al. (1981).

(12) Line 62 and 64 - define long/short distance/duration.

We have revised the terminology from "short-duration" to "short-distance" to facilitate a more precise comparison of the exercise protocols employed in the studies by Janal et al. (1984) and Haier et al. (1981). Specifically, the long-distance run conducted by Janal et al. (1984) spanned 6.3 miles (10.3 km), while the short-distance run executed by Haier et al. (1981) covered 1 mile (1.6 km).

(13) Line 62 - what type of pain?

Janal et al. (1984) implemented thermal, ischemic, and cold pressor pain in their study and observed a hypoalgesic effect in response to thermal and ischemic pain that was reversed under NLX administration. We have now specified this in the text.

(14) Line 67 - please place "i.e., the insula, ACC and prefrontal regions" in parentheses.

Done.

(15) Lines 67-69 - please provide clarity on the nature of the interventions being employed. For example, are you referring to interventions to reduce/overcome pain? Or are you referring to approaches to experimentally induce or increase pain during exercise? In either case, please be specific on the interventions employed, and why this variation in approach may make it challenging to draw a conclusion

The interventions employed by several studies aimed to investigate the pharmacological underpinnings of the pain modulatory effect of exercise and were, thus, pharmacological interventions. The primary objective of these interventions is usually not to reduce/induce/decrease/increase pain but to block a specific receptor type to infer the involvement/role of these receptor types in pain modulation through exercise. In the context of exercise and pain specifically, the most frequently used pharmacological intervention consists of administering a µ-opioid receptor antagonist (naltrexone/naloxone (NLX)). Depending on which type of µ-opioid receptor antagonist is used, different administration protocols are employed (i.e., oral or intravenous administration, different doses, only bolus without constant injection). This variability in the administration protocols of these pharmacological interventions can account for different findings of the extent of opioidergic involvement in exercise-induced pain modulation. We have now refined the according section to increase the precision and clarity of the interventions used.

(16) Line 69 - administration of what?

This passage refers to the variability of administration of µ-opioid receptor antagonists such as naloxone (NLX) or naltrexone. We have now specified this in the according line.

(17) Line 74 - EIH?

As described above, we have chosen the term 'exercise-induced pain modulation' as an umbrella term for both exercise-induced hypoalgesia and hyperalgesia. However, the reviewer is correct that specifically studies investigating exercise-induced hypoalgesia have been criticised. Still, the proposed criticism also applies to studies detecting hyperalgesia and we would, thus, argue to retain the term ‘exercise-induced pain modulation’ here for the sake of consistency.

(18) Line 75 - please define "single-arm pre-post measurements"

We appreciate the reviewers' comment. Single-arm pre-post measurement studies involve participants being assigned to a single experimental condition, with pain assessments conducted only once prior to and once following the intervention. This study design presents several limitations, particularly in the context of examining exercise-induced modulation of pain (Vaegter and Jones, 2020). Such designs do not consider the effects of habituation to noxious stimuli, as highlighted by Vaegter and Jones (2020). Consequently, when measuring pain levels with only one pre- and one post-intervention assessment, there is a risk of misinterpreting the outcomes where a reduction in post-intervention pain ratings might erroneously be credited to the exercise intervention itself, rather than being a result of habituation to the noxious stimuli experienced. Incorporating randomised controlled trials with multiple measurement blocks not only mitigates these limitations but also provides a clearer understanding of how individual bouts of exercise influence pain perception.

(19) Line 84 - is (40) a reference?

We apologise for this oversight and have now updated the reference by Borszcz et al. (2018) to be displayed correctly.

(20) Line 86 - is that 10 min per block (i.e. 40 min exercise time), or 10 min in total? If the former please include "per block" at the end of the sentence (Line 87).

The reviewer is correct in assuming that we employed 10 min of cycling per block, resulting in a total of 40 minutes of cycling. We have updated the sentence now including ‘per block’ as suggested by the reviewer.

(21) Line 89 - when you refer to "painfulness" are you referring to the intensity of pain experienced? If so, I think "pain intensity" would be more appropriate.

In the current study, participants were asked about the ‘painfulness’ of each stimulus based on previous studies (Horing et al., 2019; Horing & Büchel, 2022; Tinnermann et al., 2022). The term ‘painfulness’ is a composite measure of ‘pain intensity’ (sensory dimension) and ‘pain unpleasantness’ (affective dimension) (Talbot et al., 2019). Since unpleasantness is also a definitional criterion of pain (‘Terminology | International Association for the Study of Pain’, n.d.) and previous research shows a high correlation between ‘pain unpleasantness’ and ‘pain intensity’ (Granot et al., 2008; Talbot et al., 2019) we have opted for the term ‘painfulness’ as a more comprehensive measure. Inherently, these two measures are highly correlated.

(22) Line 91-93 - the way this is written could be suggestive of this being separate to the cycling blocks. Please rephrase to confirm that this was administered prior to the commencement of the cycling blocks.

We have refined the sentence to make it clearer that the drug treatment was administered before the cycling block commenced on each of the experimental days. We would like to further specify, that whilst the bolus dose of the treatment was administered prior to the experiment, a constant intravenous supply of SAL/NLX was maintained throughout the experiment using an infusion pump.

(23) Methods general - why only 10 min of exercise? It is likely that there is a 'dose effect' of exercise on EIH, whereby the intensity of exercise and the duration of the exercise are important. Short-duration but high-intensity exercise can induce EIH, as can moderate duration low-intensity exercise. But, for this protocol, was the intensity high enough or long enough to meet the 'dose' needed?

We thank the reviewer for their question. Our decision to employ 10-minute exercise blocks was rooted in both scientific evidence on exercise-induced hypoalgesia and the (clinical) applicability of the findings. Research has shown that exercise durations ranging from 8 minutes to 2 hours of aerobic exercise can induce hypoalgesia (for review see Koltyn (2002)). Specifically, several studies induce hypoalgesia at 10-15 minutes of aerobic exercise (Gomolka et al., 2019; Gurevich et al., 1994; Haier et al., 1981; Jones et al., 2019; Sternberg et al., 2001; Vaegter et al., 2015). Furthermore, many prior studies have employed exercise durations that are tailored to professional or amateur athletes which may not be practical for healthy individuals with lower fitness levels who may find it challenging to engage in longer sessions, such as an hour of running. When considering applying these findings to the clinical chronic pain population it is crucial to assess the manageability of proposed exercise protocols. We believe that 10 minutes of exercise, whilst being a relatively brief exercise duration, may still be sufficient to elicit exercise-induced hypoalgesia.

(24) Methods general - what was the time gap between each round (i.e. after the fMRI, how long before the participant started the next cycling block?).

After each fMRI run the participants were taken out of the MR scanner. The HR and SPO2 were measured and participants were given the chance to go to the restroom before positioning them on the bike and starting the next block. All in all, the time following the fMRI scan and before the new block commenced ranged between 5-10 minutes. We have now included this specification in the methods section.

(25) Methods general - there is some evidence to show that the EIH effect is less consistently shown when heat is used to induce pain - was there a reason heat was used as the pain induction method here?

We thank the reviewer for their comment. Indeed, previous meta-analyses by Naugle et al. (2012) report larger effect sizes for pressure pain (Cohen’s d = 0.69) closely followed by heat pain (d = 0.59). In light of this evidence, we included both pain modalities in the current study. Notably, we found no significant differences in pressure pain responses between LI and HI exercise. It is important to emphasise that the term "pressure pain" predominantly encompasses studies employing handheld pressure algometry, whereas our investigation utilised a pressure cuff. This methodological variation raises the possibility that our findings—and corresponding effect sizes—may not be directly comparable to prior pressure pain studies.

(26) Methods general - please be consistent in the use of terminology. In some areas, you use the phrase "cycling block" whereas in other areas it is referred to as a "cycling run".

We have revised the methods section to be more precise with the terms ‘run’ and ‘block’.

(27) Line 571-573 - Please detail how participants were excluded based on scores from STAI and BDI-II.

We apologise for the misspelling, as it should be that one participant was excluded based on a BMI (body mass index) below 18. No participant had to be excluded based on the STAI or BDI-II score in the current study. We have corrected this in the manuscript.

(28) Line 636-651 - the FTP20 test has been shown not to be a valid marker of the separation between the heavy and severe exercise intensity domains (see Wong et al 2023 - https://doi.org/10.1080/02640414.2023.2176045). Given that participants completed the high intensity cycle in 'zone 4' (91-106% of FTP), it is probable that participants could have completed this 10 min in either the heavy or the severe exercise intensity domains, with significant implications for the relative challenge this 10 min of exercise. Why was zone 4 used? What are the implications of this? Please discuss and include this as a limitation.

We thank the reviewer for their comment as it touches upon the challenges of accurately estimating exercise intensities. It is indeed crucial to consider the boundaries between moderate, heavy, and severe intensity domains, as delineated by physiological markers.

The study by Wong et al. (2023) is interesting; it assesses blood lactate and VO2 levels at FTP and FTP+15 Watts. Despite being highly relevant for the field some of the findings should be interpreted with caution due to the low sample size of 13 participants, consisting of 11 male and only 2 female cyclists, which may limit generalisability. Additionally, the testing protocol implemented in the study to determine participants' FTP consisted of a 5-minute self paced pedalling at 100 Watts followed by a 20-minute maximal, self-paced time trial. This differs from the FTP20 test as implemented in the current study (see Supplemental Table S1) or by other studies (McGrath et al., 2019). The finding in Wong et al. (2023) that participants were only able to sustain cycling at FTP for an average of 33 minutes suggests that the deviating protocol overestimates FTP. Mackey and Horner (2021) propose that the validity of the FTP20 test might rely on the warm-up used before FTP20 testing and the training status of athletes.

However, we acknowledge that without direct measurements of VO2max or blood lactate levels, it is challenging to determine the precise intensity domain in which each participant was operating in the current study. Still, the RPE (low: M = 8.59, SD = 1.32; high: M = 14.92, SD = 1.98) suggests that participants operated in the heavy-intensity domain in the HI exercise condition. This is further supported by the relative power (%FTP) maintained in the HI (M = 105; SD = 0.05; Author response image 5, purple) and LI (M = 58; SD = 0.06; Author response image 5, green) exercise conditions (difference: t(37) = 44.58, P < 2.2e-16, d = 6.46) confirming the accuracy of the implemented FTP test as well as the maintained power throughout the cycling blocks. Thus, we would argue that participants in the current study predominantly exercised the heavy domain during the HI exercise condition. We have included the relative Power in Figure 3A, replacing the absolute Power.

Finally, we propose that discussing exercise intensity domains within the context of our study enriches the understanding of exercise-induced hypoalgesia without undermining the integrity of our findings. We have now included a discussion of the validity of the FTP20 test as a demarcation point concerning the intensity domains.

**Author response image 5. sa3fig5:** Raincloud plot of relative power (%FTP) during low (green) and high (purple) intensity exercise. Individual data points depict subject-specific averages across blocks.

(29) Line 676 - please provide further information on each cycling run/block. Did each participant complete a total of 4 runs (i.e., a total of 40 minutes of exercise), with 2 runs completed at a high intensity and 2 runs completed at a low intensity in a randomised order (e.g., for one participant this could be 10 minutes at low, followed by 10 minutes at high, followed by 10 minutes a low, followed by 10 minutes at high)? Figure 1 details this nicely, however, it would be helpful to read in-text.

The reviewer is correct in assuming that there were a total of 4 blocks on each experimental day. Participants completed cycling in 2 blocks at HI and in 2 blocks at LI in a pseudorandomised order. This order was kept constant across experimental days (i.e. completing the same block order on Day 2 and Day 3). We have detailed this further in the Methods section.

(30) Discussion general - it is possible that EIH could be induced via different mechanisms and that these mechanisms are at least in part due to exercise intensity. For example, EIH from higher-intensity exercise might have some contribution from CPM.

We thank the reviewer for their comment. Previous research aimed to disentangle the two seemingly similar mechanisms of exercise-induced hypoalgesia (EIH) and conditioned pain modulation (CPM) (Ellingson et al., 2014; Rice et al., 2019; Samuelly-Leichtag et al., 2018; Vaegter et al., 2014). CPM is typically induced by applying a tonic noxious stimulus that decreases pain sensitivity to another noxious stimulus applied simultaneously or shortly after at a distant body part (Graven-Nielsen & Arendt-Nielsen, 2010). Despite EIH and CPM showing distinct mechanisms, it cannot be completely ruled out that there are at least partially overlapping mechanisms driving the two phenomena (Rice et al., 2019). Due to our study design, where the time difference between cycling blocks and the applied pain was on average five minutes, it is unlikely that CPM is the driving pain modulatory mechanism in our study setup.

(31) Line 101 - as this was preregistered, should the study design be followed and then reported?

We have conducted the study adhering to the preregistered study design and now report the results for pressure pain (Supplemental Figure S1). Some of the preregistered analyses (i.e. directly comparing heat and pressure pain) were beyond the scope of the current study and will be reported separately.

(32) Line 110 - please provide some data on the fitness levels and how this is classified as high/low.

The FTP (relative to body weight) was used as an estimate of cardiovascular and endurance fitness (Valenzuela et al., 2018). We refrained from classifying the fitness levels dichotomously as low or high since this is a subjective measure in a sample of healthy individuals of diverse fitness levels. Instead, we utilised the FTP as a more nuanced metric for comparison.

(33) Lines 159-160 - in the context of the difference in intensity between the sessions. But, it is likely that the high-intensity exercise would have posed quite different relative challenge between participants.

We thank the reviewer for their comment. As described above, we did not obtain direct measurements of VO2max or blood lactate levels making it challenging to determine the precise intensity domain in which each participant was operating in the current study. However, all participants received the same instructions to the BORG rating scale ensuring the comparability of RPE across participants to a certain extent.

(34) Figure 3C - what instructions and familiarisation were given to participants regarding the 6-20 Borg scale? In Figure 3C it looks as though several participants rated the low exercise intensity at 6. This would/should be equivalent to sitting quietly, so it looks as though at least several participants did not understand how to use the RPE - please discuss.

Indeed, three participants rated the LI exercise condition at 6 due to an error in the translation of the scale instruction. Participants were instructed that the lower anchor point of the scale (6) referred to ‘extremely light’ instead of ‘no exertion’. Thus, we have rescaled the RPE ratings where a rating of 6 now corresponds to a 7 (‘extremely light’) on the BORG scale and again calculated the paired t-test. There is still a significant difference in the RPE between exercise intensities (t(38) = 19.65, P < 2.2e-16, d = 3.69; Author response image 6). We have corrected this in the manuscript accordingly and updated Figure 3C.

**Author response image 6. sa3fig6:** Raincloud plot of rating of perceived exertion (RPE) on the BORG scale during low (green) and high (purple) intensity exercise. Individual data points depict subject-specific averages across blocks. A rating of 6 reflects ‘no exertion’ and 20 reflects ‘maximal exertion’.

(35) Line 171 - is (37, 38) a reference?

We apologise for this oversight and have now updated the references to be displayed correctly.

(36) Line 176-18 - is this interaction sufficiently powered? Differences between sexes are not mentioned in the pre-registered study

We have conducted an additional post-hoc power analysis for the interaction of drug, fitness level, and sex on differential heat pain ratings. We employed the power analysis for mixed models implemented in R (powerCurve) with 1000 simulations. This revealed that with a power of α = 0.8, a sample size of n = 27 would have been sufficient to detect this effect (Author response image 7). Despite not having preregistered the factor ‘sex’, we believe that the observed results provide valuable insights that contribute to a deeper understanding of the data. We have established these analyses to be exploratory, emphasising the need for caution in their interpretation. However, we feel it is essential to report these findings to inform future studies, ensuring that such factors are adequately considered.

**Author response image 7. sa3fig7:** Post-hoc power analysis for behavioural effects from the linear mixed effects (LMER) model with interaction drug, fitness level, and sex using the R package powerCurve with α = 0.8 and 1000 simulations.

(37) Line 227 - this is not what this analysis shows. The comparison is low vs high-intensity exercise on pain modulation, not exercise vs. no exercise. You cannot conclude that aerobic exercise has no effect on pain modulation because you did not do that comparison (i.e. no baseline (without exercise) for pain).

We agree with the reviewer and have rephrased the sub-headline accordingly to reflect that there is no difference in exercise-induced hypoalgesia between HI and LI aerobic exercise.

(38) Methods General - why was a control condition not used, or at least a baseline pain response, so that low/high-intensity exercise could be compared to a baseline? Given this, I'm not sure I agree with the study conclusions (abstract: 'These results indicate that aerobic exercise has no overall effect on pain in a mixed population sample') because you have compared high vs low-intensity exercise, not exercise vs. no exercise.

As for the lack of a resting control condition, we acknowledge that our study was not designed to test the overall effect of exercise versus no exercise. However, our primary objective was to compare different exercise intensities, hypothesising that low-intensity (LI) exercise would induce less pain modulation as compared to high-intensity (HI) exercise. By exploring this, we aimed to enhance understanding of the dose-response relationship between exercise and pain modulation. To better reflect this focus, we have revised the misleading phrasing regarding the ‘overall’ effect of exercise to clearly emphasize our primary aim: comparing HI and LI exercise. This reviewer suggests an interesting interpretation of the data suggesting that exercise-induced hypoalgesia might have occurred for both exercise intensities since the pain ratings provided were lower than the anticipated intensities as determined by the calibration. Given that this difference is lower in the naloxone (NLX) condition could provide evidence of opioidergic mechanisms underlying this effect.

Unfortunately, the current study is not designed to comprehensively answer this question since there was no resting control condition. In particular, the lower pain ratings under SAL (Figure 6) could be due to exercise triggering the descending pain modulatory system (DPMS), but equally due to the default activation of the DPMS. Only an additional “no exercise” condition could disentangle this. Furthermore, habituation to noxious stimuli can influence pain ratings, resulting in lower pain ratings during the experiment as compared to the calibration.

(39) Line 285 - or that better-trained individuals have a greater EIH response to higher intensity exercise, but both those of low and high fitness have established EIH after low intensity exercise. Given there isn't a 'no exercise' baseline, it is hard to make conclusions about EIH effect generally, only comparisons between high/low exercise intensity.

We thank the reviewer for their comment. We agree that we cannot establish whether all participants showed a hypoalgesic response to the LI exercise with the current study design. However, our results show that participants with higher fitness levels showed increased hypoalgesia after HI exercise compared to those with lower fitness levels. We have refined the sentence accordingly.

(40) Figure 7A - the regression line here is not that convincing.

We acknowledge the reviewers’ concern regarding the regression line. However, it is important to note that the significant main effect of fitness level on differences in pain ratings in the SAL condition (β = 6.45, CI [1.25, 11.65], SE = 2.56, t(38) = 2.52, P = 0.02) supports the assertion that higher fitness levels are associated with greater hypoalgesia following HI exercise compared to LI exercise. While the trend may not be visible for all data points, the statistical analysis provides a robust basis for the observed relationship (r = 0.33, P = 0.038).

(41) Line 354 - the NLX infusion was double-blind, but what are the implications of participants knowing that they completed high/low-intensity exercise - this cannot be blinded.

The reviewer is correct that the exercise intensities cannot be blinded. To account for potential expectation effects of exercise on several psychological and physiological domains (including pain), participants completed a questionnaire on the calibration day where they had to indicate their expectations of to what extent acute exercise affects several domains (Lindheimer et al., 2019). They could rate each domain on a Likert scale ranging from ‘large decrease’ (-3) to ‘large increase’ (3) with 0 denoting ‘no effect’. This format was chosen to allow measuring the direction and magnitude of expectation effects and to avoid being directive or suggestive (Lindheimer et al., 2019). Despite including other psychological and physiological domains in the questionnaire (i.e., stress, anxiety, energy, memory) we focused on the specific pain domains (muscle pain, joint pain, and whole body pain) to establish participant’s expectations regarding the effect of acute exercise on pain. We tested whether the expectation ratings for each pain type were significantly different from 0 (no effect) using a one-sample t-test.

There was no significant effect for muscle pain (t(38) = 1.78, P = 0.08, M = 0.39, SE = 0.12), joint pain (t(38) = -0.12, P = 0.90, M = -0.03, SE = 0.11), or ‘whole-body pain (t(38) = -1.05, P = 0.30, M = -0.21, SE = 0.12) suggesting there to be no expectation effect on these pain domains in the overall sample (Supplemental Figure S10A). Since there is variation in the data we calculated the correlation of the expectation ratings in the different pain domains with the difference score between the pain ratings in the SAL condition (LI – HI rating; Supplemental Figure S10B). This analysis yielded no significant correlation in either of the pain domains (joint pain: r = 0.11, P = 0.49; muscle pain: r = -0.07, P = 0.68; whole-body pain: r = 0.07, P = 0.68).

Moreover, given that we have not been able to show a difference between the exercise intensities on pain modulation, expectation effects are likely not to contribute to this null effect.

(42) Line 356-358 - and this comparison (and primary hypothesis) is not blinded.

While we agree with the reviewer that this comparison is not – and potentially cannot be – blinded, we would like to reiterate our results from the previous paragraph that indicate that such expectation effects of exercise on pain were not present in the sample and, thus, did not seem to have influenced the results. It is noteworthy that the double-blind design of our study design specifically pertains to the pharmacological intervention employed.

(43) Line 358-360 - this could be explained by both types of exercise inducing EIH via the same mechanism (which is disrupted by NLX).

We thank the reviewer for their comment and would like to refer back to the reviewer's comment number 38 for a response to this.

(44) Line 360-361 - this conclusion cannot be drawn, because you have only compared high vs low intensity exercise. So, the conclusion should be 'These results suggest that there is no difference between high and low aerobic exercise intensity on heat-induced pain'.

We agree with the reviewer and have rephrased the sentence to reflect the claim accurately.

(45) Line 396 - as previously discussed, this conclusion cannot be drawn through this study design.

We agree with the reviewer and have rephrased the sub-headline accordingly to reflect that there is no difference in exercise-induced hypoalgesia between HI and LI aerobic exercise.

(46) Line 399 - please expand on this point - it is critical to the hypothesis and should also be included in the introduction. What intensities/duration/dose of aerobic exercise is generally established to cause EIH?

We thank the reviewer and agree that this is a crucial aspect that requires further specification. Below we have expanded on the duration/intensities shown to elicit exercise-induced hypoalgesia and included a concise version of this detailed paragraph in the manuscript introduction.

For aerobic exercise, different methods have been employed to determine exercise intensity levels i.e., through the VO2max, age-predicted HRmax, or incremental intensities (Koltyn, 2002). Most studies using VO2max as a measure of exercise intensity (Koltyn et al., 1996; Micalos & Arendt-Nielsen, 2016; Vaegter et al., 2014) were able to induce hypoalgesia with HI levels ranging between 65%-75% VO2max. When using the HRmax as a measure of determining exercise intensities, HI exercise at 70%-75% of the HRmax has been shown to produce greater hypoalgesia compared to moderate intensity at 50% HRmax (Naugle et al., 2014; Vaegter et al., 2014). Furthermore, previous research has suggested that HI exercise produces greater hypoalgesia compared to LI exercise (60-70% HRmax vs. light activity: M. D. Jones et al., 2019; 70% vs. 50% HRmax: Naugle et al., 2014; 75% vs. 50% VO2max: Vaegter et al., 2014).

Furthermore, different durations can be regarded as suitable with durations between 8 minutes to 2 hours of aerobic exercise having been shown to induce hypoalgesia (for review see Koltyn (2002)). Hoffman et al. (2004) showed a hypoalgesic response after 30 minutes but not after 10 minutes at 75% VO2max of cycling. In contrast, other studies were able to induce hypoalgesia at 10-15 minutes of HI aerobic exercise (75% VO2may: Gomolka et al., 2019; 63% VO2max: Gurevich et al., 1994; self-paced: Haier et al., 1981; 60-70% HRmax: Jones et al., 2019; 85% HRmax: Sternberg et al., 2001; 75% VO2max: Vaegter et al., 2015).

(47) Line 400-401 - please define high intensity.

We thank the reviewer for their comment. The referenced studies by Vaegter et al. (2014) and Jones et al. (2019) based the estimation of HI and LI exercise on an age-related target heart rate corresponding to VO2max and HRmax, respectively. In Vaegter et al. (2014), the HI condition corresponded to 75% VO2max, while the LI to 50% VO2max. In Jones et al. (2019), the HI exercise condition corresponded to 60% and 70% of HRmax, while the LI condition was defined as pedalling slowly against a light resistance of 0.5 kg of force to maintain a rating of perceived exertion (RPE) not above resting. We have included this clarification in the relevant section to elucidate the intensities of the chosen exercise conditions.

(48) Line 403-405 - I'm not sure I follow (perhaps I have misunderstood) - pain induction was completed after exercise in the MRI scanner, so there was no distraction effect of exercise in either condition. A baseline could have been established in the same way and there would be exactly the same conditions, just without prior exercise.

We agree with the reviewer that a resting baseline condition in the context of exercise induced pain modulation allows for the investigation of a potential hypoalgesic effect of exercise compared to no exercise. Nevertheless, it is important to note that previous studies (Brooks et al., 2017; Sprenger et al., 2012) have shown that cognitive pain modulation is mediated by endogenous opioids. Therefore, tasks with different attentional loads potentially influence post-task pain ratings. Although, we agree with the reviewer that the effect of distraction or attentional load would be minimal in the MR scanner, there still could be an effect of different cognitive loads from exercise vs. no exercise. Nevertheless, we focus the discussion on investigating the dose-response relationship between different exercise intensities where an ‘active’ control condition might contribute to a more nuanced understanding of exercise-induced pain modulation.

(49) Line 403-411 - this is fine (although I do not agree that this was the best methodological decision), however, it does limit the conclusions that can be drawn (as previously mentioned). That is, you cannot conclude that no EIH occurred, only that there was no difference between low and high-intensity exercise in post-exercise pain response.

We agree with the reviewer that the comparison of HI vs. LI exercise does not allow for an interpretation of the overall effect of exercise as opposed to no exercise on pain modulation. The comparison of HI and LI exercise allows the investigation of a dose-response relationship of these distinct exercise intensities. While LI exercise might not be a 'pure' control condition in the traditional sense, it is valuable for exploring the complexities of exercise and pain interaction.

(50) Line 419-422 - sorry I do not follow - you say that moderate intensity exercise most reliably induces EIH but then select exercise intensities that are likely to be in the heavy or severe intensity domain? Please also include in this discussion the limitations of FTP20 as a threshold marker (see Wong et al) and the implications on the results/conclusions.

We thank the reviewer for their comment. In the referenced sentence, we have defined the HI exercise as described in the reviews. Specifically, Wewege and Jones (2020) reported hypoalgesia to be greater after higher-intensity exercise, although the intensity was not further specified. Naugle et al. (2012) noted that HI exercise (i.e., 75% of VO2max) produced greater hypoalgesia, while Koltyn (2002) indicated that hypoalgesia occurs at intensities ranging from 60% to 75% of VO2max but more reliably at 75% VO2max or higher. Consequently, we have removed the term ‘moderate’, as it does not accurately reflect what has been reported in the reviews and could be misleading. Moreover, we have clarified the specific criteria for what is considered high (or higher) intensity exercise in the referenced reviews.

We kindly ask the reviewers to refer back to the previous comment (reviewer comment number 28) regarding the discussion of the intensity domains and the FTP20 test as demarcation point for these intensity domains.

(51) Line 422-425 - indeed, pacing is an important element of this test, which inexperienced cyclists have difficulty with when they are not provided with proper familiarisation.

We agree with the reviewer that the FTP20 test has mainly been validated and employed in experienced cyclists and requires further validation in non-athletes of both sexes. However, since we have used an extensive warm-up period and several paced steps (intervals, 5-minute time-trial) as well as recovery periods (Supplemental Table S1) based on McGrath et al. (2019) we propose that participants were thoroughly familiarised with the elements of pacing before the estimation of the FTP in the 20-minutes took place. On average, participants showed a variation of M = 21.80 Watts (SE = 1.44 Watts) during the 20-minute paced FTP20 test (Supplemental Figure S11A). Interestingly, our data suggests that participants with a higher FTP showed higher variation of power output (Watts) during the 20-minute FTP test compared to individuals with lower fitness levels (Supplemental Figure S11B).

(52) Line 425-427 - please remove this, the RPE difference between exercise bouts is not evidence that participants cycled at FTP.

We thank the reviewer for their comment. However, we would propose to include the rating of perceived exertion (RPE) since it shows that the exercise intensities have been perceived as significantly different by the participants. This behavioural measure of exertion is potentially important for a broader audience to understand the exercise implementation beyond physiological markers.

(53) Line 432 - high vs. low-intensity aerobic exercise

We have changed the sentence accordingly to support the claim of the study that there was no difference in exercise-induced pain modulation between HI and LI aerobic exercise.

(54) Line 447-449 - this seems contradictory to the first line of this paragraph (430-432) - i.e. that the heterogenous sample may have caused the null finding. Why deliberately select a participant sample that is likely to lead to a null effect?

In the current study, we aimed to include participants of diverse fitness levels and both sexes to verify the findings on exercise-induced pain modulation in a broader population. We consider this important concerning translational aspects of EIH. Indeed, our heterogeneous sample may have ‘caused’ the observed null effect, but at the same time, it suggests that more homogenous (sometimes composed solely of male athletes) samples employed in many earlier studies might have skewed the understanding of exercise-induced pain modulation and thus unintentionally suggested a (non-existing) generalisation of this effect to the general population.

(55) Line 532-456 - although Koltyn found electrical pain to have the greatest effect?

The review by Naugle et al. (2012) reported effect sizes for heat (Cohens d = 0.59) and pressure pain intensity (d = 0.69) following aerobic exercise but did not provide effect sizes for electrical pain intensity. They noted that the effect size for electrical pain intensity after isometric exercise was d = 0.40, which is lower than that for heat and pressure pain. While Koltyn (2002) stated that electrical and pressure stimuli induce exercise-induced hypoalgesia more consistently than thermal pain, the study did not clarify whether this applies to pain threshold, intensity, or tolerance, nor did they provide effect sizes. Given that electrical, pressure, and heat pain are the most commonly used methods to induce quantifiable pain in the context of exercise studies (Vaegter and Jones, 2020), we based our decision to use heat and pressure pain primarily on Naugle et al.'s findings.

(56) Line 468-469 - why leave out content that was pre-registered (i.e. difference between pressure and heat pain) but includes analysis that wasn't (i.e. sex differences)? If a study is going to be pre-registered, then isn't it important to follow that design?

We thank the reviewer for this comment. We have conducted the study adhering to the preregistered study design and now report the results for pressure pain (Supplemental Figure S1). Some of the preregistered analyses (i.e. directly comparing heat and pressure pain) were beyond the scope of the current study and will be reported separately.

(57) Line 532-525 - and how could this have been accounted for?

We apologise for any confusion, as we are unsure about the specific reference the reviewer is making based on the provided line numbers. We believe the question relates to how the potential effects of endocannabinoids were considered in the current study design, and we've addressed that in our response. In human studies, it is not possible to centrally block endocannabinoids, which makes it difficult to directly estimate their role in exercise-induced pain modulation in humans. Measuring endocannabinoids in the blood might not adequately capture changes in endocannabinoid levels in the brain throughout the different exercise intensity conditions. Despite these limitations, exploring the role of endocannabinoids in exercise-induced pain modulation presents a promising avenue for future research that could enhance our understanding of pain mechanisms and improve pain management strategies.

1. Limitations General - please include the other limitations discussed in this review.

Done.

(59)Line 530 - please amend this conclusion, in line with previous comments.

Done.

We would like to thank the reviewer for critically evaluating the manuscript and providing insightful comments. We appreciate the reviewer recognising the strengths of our work and believe that their suggestions will contribute to improving the quality of the manuscript.